# PHLPP1 counter-regulates STAT1-mediated inflammatory signaling

**Ksenya Cohen Katsenelson[1†], Joshua D Stender[2†], Agnieszka T Kawashima[1,3†], Gema Lordén[1], Satoshi Uchiyama[4], Victor Nizet[4,5], Christopher K Glass[2], Alexandra C Newton[1]\***

[1]Department of Pharmacology, University of California, San Diego, San Diego, United States; [2]Department of Cellular and Molecular Medicine, University of California, San Diego, San Diego, United States; [3]Department of Pharmacology and Biomedical Sciences Graduate Program, University of California, San Diego, San Diego, United States; [4]Department of Pediatrics, University of California, San Diego, San Diego, United States; [5]Skaggs School of Pharmacy and Pharmaceutical Sciences, University of California, San Diego, San Diego, United States

**Abstract** Inflammation is an essential aspect of innate immunity but also contributes to diverse human diseases. Although much is known about the kinases that control inflammatory signaling, less is known about the opposing phosphatases. Here we report that deletion of the gene encoding PH domain Leucine-rich repeat Protein Phosphatase 1 (PHLPP1) protects mice from lethal lipopolysaccharide (LPS) challenge and live *Escherichia coli* infection. Investigation of PHLPP1 function in macrophages reveals that it controls the magnitude and duration of inflammatory signaling by dephosphorylating the transcription factor STAT1 on Ser727 to inhibit its activity, reduce its promoter residency, and reduce the expression of target genes involved in innate immunity and cytokine signaling. This previously undescribed function of PHLPP1 depends on a bipartite nuclear localization signal in its unique N-terminal extension. Our data support a model in which nuclear PHLPP1 dephosphorylates STAT1 to control the magnitude and duration of inflammatory signaling in macrophages.
DOI: https://doi.org/10.7554/eLife.48609.001

**\*For correspondence:**
anewton@ucsd.edu

[†]These authors contributed equally to this work

**Competing interests:** The authors declare that no competing interests exist.

## Introduction

Gene expression is an exquisitely regulated process that maintains cellular homeostasis and orchestrates appropriate responses to environmental stimuli such as hormones, cytokines, and pathogenic microbes (*Dawson and Kouzarides, 2012*; *Flavahan et al., 2017*). Homeostatic control of inflammatory genes is particularly relevant to cancer since chronic inflammation promotes tumorigenesis and influences patient response to cancer therapeutics (*Coussens and Werb, 2002*; *Grivennikov et al., 2010*). Dysregulated gene expression, a hallmark of cancer, can arise from mutations in transcription factors (exemplified by p53; see *Sabapathy and Lane, 2018*), alterations in signaling pathways controlling transcription factor function (for example, hormone-dependent transcription factors in prostate and breast cancers; see *Jernberg et al., 2017*; *Pejerrey et al., 2018*), or upregulation of oncogenic transcription factors (notably c-myc, which regulates essential cell-cycle checkpoints; see *Kalkat et al., 2017*). Aberrant protein phosphorylation underpins all of these mechanisms, via dysregulation of signaling pathways, alterations in transcription factor machinery, and/or effects on the chromatin epigenetic landscape (*Rossetto et al., 2012*; *Whitmarsh and Davis, 2000*). Thus, targeting phosphorylation mechanisms is of considerable therapeutic interest.

Macrophages are among the first responders to infection, engaging foreign pathogens via pattern recognition receptors, including the Toll-like receptors (TLRs). TLRs are a conserved family of

cell surface or phagosome-associated receptors that discriminate distinct features of microbial and viral pathogens, including lipoproteins (TLR1/2/6), lipopolysaccharide (LPS) (TLR4), flagellin (TLR5), single-stranded RNA (TLR7/8), double-stranded RNA (TLR3), and double-stranded DNA (TLR9) (*Karin et al., 2006*; *O'Neill et al., 2013*). Upon pathogen recognition by TLRs, a pro-inflammatory response is initiated that activates the signal-dependent transcription factors nuclear factor-κ B (NFκB), activator protein 1 (AP1), interferon response factors (IRFs), and, through secondary mechanisms, the signal transducer and activator of transcription (STAT) protein family (*O'Neill et al., 2013*). These activated transcription factors function in a combinatorial manner to drive expression of antimicrobial and inflammatory response genes that aid in elimination of foreign pathogens. However, while inflammation is required for protection against foreign microbes, it can lead to excessive cytokine production, chronic inflammation, and cancer if not properly resolved (*Coussens and Werb, 2002*; *Fullerton and Gilroy, 2016*; *Grivennikov et al., 2010*). Thus, macrophages have evolved regulatory mechanisms to resolve inflammatory responses in a timely manner, including shut down of STAT1 signaling pathways by the suppressor of cytokine signaling (SOCS) family of proteins (*O'Shea and Murray, 2008*), suppression of nitric oxide production by the enzyme arginase (*Wynn and Vannella, 2016*), and inhibition of a key subset of NFκB-dependent genes by anti-inflammatory omega-3 fatty acids (*Oishi et al., 2017*).

STAT1 is the founding member of the STAT transcription factor family and serves as a paradigm for how phosphorylation regulates transcription factor structure, function, and localization (*Darnell et al., 1994*; *Morris et al., 2018*; *Stark and Darnell, 2012*). In the canonical pathway, STATs are recruited from the cytosol to cytokine-bound and Tyr-phosphorylated receptors where they are phosphorylated on a key Tyr residue (Tyr701 for STAT1) by Janus Kinases (JAKs). This phosphorylation event promotes STAT dimerization and nuclear entry, allowing STAT binding to specific promoter sequences and thus initiating gene transcription. Upon promoter binding, STATs become additionally phosphorylated on a regulatory Ser residue at a MAPK consensus sequence (Ser727 for STAT1), a modification that enhances their transcriptional activity (*Darnell, 1997*; *Sadzak et al., 2008*; *Wen et al., 1995b*; *Whitmarsh and Davis, 2000*). Importantly, STAT1 transduces signals from type I and II interferons (IFNs), resulting in binding to IFN-stimulated response elements (ISREs) and to IFN-gamma (IFNγ)-activated site (GAS) elements in the promoters of IFN-stimulated genes (ISGs), inducing their transcription and stimulating inflammation (*Platanias, 2005*). While the kinases that phosphorylate Tyr701 and Ser727 on STAT1 have been extensively studied, as have been the phosphatases that dephosphorylate Tyr701, the phosphatases that oppose the Ser727 phosphorylation are unknown.

PH domain Leucine-rich repeat Protein Phosphatase 1 (PHLPP1) is one of the newest members of the phosphatome (*Chen et al., 2017*; *Gao et al., 2005*). Originally discovered for its function in suppressing growth factor signaling by dephosphorylating Akt on the hydrophobic motif site, Ser473 (*Gao et al., 2005*), the repertoire of PHLPP1 substrates is continually expanding (*Grzechnik and Newton, 2016*). PHLPP1 is a bona fide tumor suppressor: its expression is frequently lost in cancer and its genetic ablation in a mouse model results in prostate neoplasia (*Chen et al., 2011*; *Liu et al., 2009*). PHLPP1 is also involved in the immune response, where its dephosphorylation of Akt reduces the capacity of regulatory T cells to transduce T cell receptor signals, a key function in T cell development (*Patterson et al., 2011*). Additionally, mice lacking PHLPP1 have enhanced chondrocyte proliferation as a result of increased Akt2 activity, diminished FoxO1 levels, and increased *Fgf18* expression, suggesting PHLPP1 inhibition could be a strategy to promote cartilage regeneration and repair (*Bradley et al., 2015*). PHLPP1 also suppresses receptor tyrosine kinase gene expression by a mechanism distinct from its effects on Akt, to influence growth factor signaling, including that mediated by the epidermal growth factor (EGF) receptor (*Reyes et al., 2014*).

PHLPP1 is unusual among protein phosphatases in that its regulatory modules and catalytic domain are on the same polypeptide. Most notably, it has a PH domain essential for dephosphorylation of protein kinase C (PKC) (*Gao et al., 2008*), a PDZ ligand necessary for Akt recognition (*Gao et al., 2005*), and a leucine-rich repeat (LRR) segment required for transcriptional regulation of receptor tyrosine kinases (*Reyes et al., 2014*). In addition, PHLPP1 possesses an approximately 50 kDa N-terminal extension (NTE) of unknown function. Stoichiometric association with substrates by direct binding to the protein-interaction domains on PHLPP or common scaffolds (e.g. PDZ domain proteins such as Scribble; see *Li et al., 2011*) allows fidelity and specificity in PHLPP function, and may account for its > 10 fold lower catalytic rate compared to the closely related phosphatase

PP2Cα (*Sierecki and Newton, 2014*). Given its transcriptional regulation of at least one family of genes (*Reyes et al., 2014*), PHLPP1 is an attractive pharmacological target for modulation of gene expression.

Here we report that nuclear-localized PHLPP1 opposes STAT1 Ser727 phosphorylation to inhibit its transcriptional activity and promote normal resolution of inflammatory signaling. We find that *Phlpp1*$^{-/-}$ mice have improved survival following infection with *Escherichia coli* (*E. coli*), indicating a role of the phosphatase in innate immunity. Since macrophages are key in the initial response to lipopolysaccharide (LPS) from Gram-negative bacteria such as *E. coli*, we further explored the role of PHLPP1 in controlling LPS-dependent signaling in this cell type. The STAT1 binding motif was identified from the most common promoter sequences of 199 genes that remained elevated following LPS treatment of bone marrow-derived macrophages (BMDMs) from *Phlpp1*$^{-/-}$ mice compared to those from wild-type (WT) mice. We validated common transcriptional targets of PHLPP1 and STAT1, showing that loss of PHLPP1 upregulates the transcription of several genes including guanylate binding protein 5 (*Gbp5*), whereas loss of STAT1 downregulates them. Cellular studies revealed that dephosphorylation of STAT1 on Ser727 suppresses its transcriptional activity by a mechanism that depends both on the catalytic activity of PHLPP1 and a previously undescribed nuclear localization signal (NLS) in the NTE of PHLPP1. Taken together, our results identify PHLPP1 as a major player in the resolution of inflammatory signaling.

## Results

### PHLPP1 regulates the innate immune response

To explore the role of PHLPP1 in acute inflammation, we examined the kinetics and outcome of sepsis-induced death caused by intraperitoneal (i.p.) injection of Gram-negative *E. coli* bacteria in WT and *Phlpp1*$^{-/-}$ mice. Surprisingly, absence of PHLPP1 provided a strong protective effect; at a dose where more than 50% of WT mice died within 12 hr of *E. coli* challenge, 50% of the *Phlpp1*$^{-/-}$ mice remained alive after 10 days (*Figure 1A*). Similarly, *Phlpp1*$^{-/-}$ mice were protected from toxicity induced by the purified Gram-negative bacterial cell wall component LPS, with nearly half of the *Phlpp1*$^{-/-}$ mice alive after 10 days compared to only 1 out of 16 of the WT mice (*Figure 1B*). To understand the lower mortality rates in *Phlpp1*$^{-/-}$ mice, we measured levels of different cytokines in the serum of mice across a time course following LPS injection (*Figure 1C–E*). Serum levels of pro-inflammatory cytokine interleukin 6 (IL-6) were significantly increased in WT mice within 5 hr of LPS injection, returning to baseline within 12 hr (*Figure 1C*). In contrast, the *Phlpp1*$^{-/-}$ mice had 2-fold lower IL-6 levels at 5 hr post-infection, but these levels were sustained for up to 24 hr, suggestive of improper resolution of inflammation. Levels of another pro-inflammatory cytokine, IL-1β, were likewise consistently higher in *Phlpp1*$^{-/-}$ mice compared with WT mice (*Figure 1D*). By contrast, levels of the anti-inflammatory cytokine IL-10 did not differ significantly between the WT and *Phlpp1*$^{-/-}$ mice (*Figure 1E*). Note that cytokine levels were measured up to 24 hr post LPS injection, when the protective effect of PHLPP1 loss was not yet apparent. These findings indicate an essential role for PHLPP1 in regulation of the innate immune response at the whole organism level.

### Loss of PHLPP1 results in increased STAT1-dependent transcription in macrophages

Since macrophages are a key cell type involved in the initial response to *E. coli* infection and LPS challenge, we analyzed the transcriptome of BMDMs isolated from WT or *Phlpp1*$^{-/-}$ mice before and after stimulation by the major LPS component, Kdo2-Lipid A (KLA), for 1, 6, or 24 hr (*Figure 2A*). RNA-Seq analysis identified 1,654 mRNA transcripts induced more than two-fold by KLA treatment, with a false discovery rate (FDR) less than 0.05 at any of the time points. Expression of approximately 12% of these genes (199 genes; *Supplementary file 1*) was increased in macrophages from *Phlpp1*$^{-/-}$ mice compared to those from littermate control WT mice 6 hr following KLA treatment; transcript levels of these genes remained significantly elevated (>two fold) 24 hr later. Another set of genes exhibited reduced expression 24 hr following KLA treatment (144 genes; *Supplementary file 2*). Gene ontology analysis revealed that many of the genes whose expression was elevated in the *Phlpp1*$^{-/-}$ macrophages are associated with inflammatory signaling: these included genes annotated for their involvement in the innate immune response, cytokine-cytokine receptor interactions, LPS

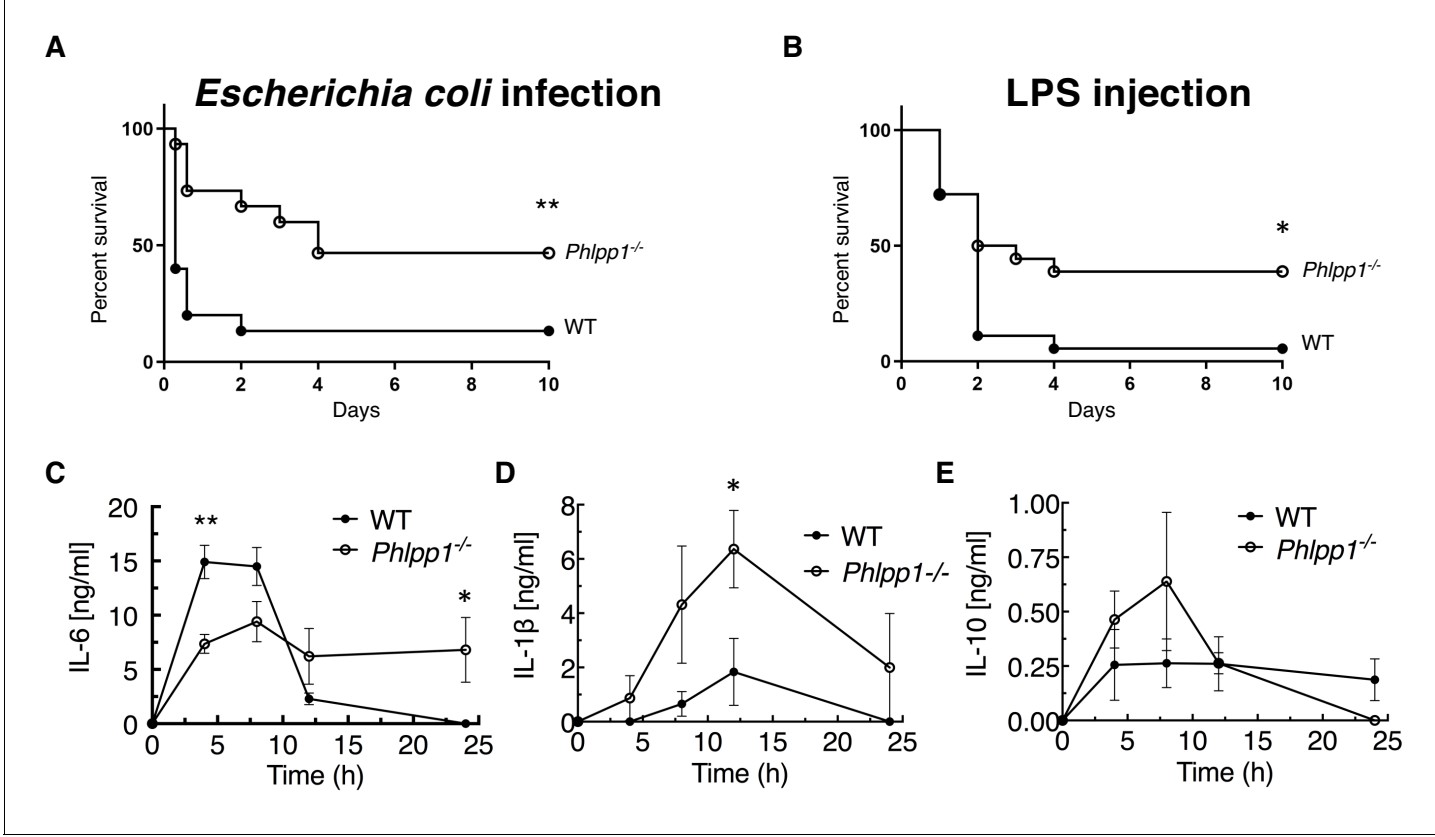

**Figure 1.** PHLPP1 knock-out mice are protected against sepsis-induced death. (**A**) Survival curve of WT and *Phlpp1*[-/-] mice i.p. infected with $1 \times 10^7$ cfu of *E. coli*. Values are expressed as percent survival of 15 mice for each genotype. \*\*p<0.01 by log-rank test. (**B**) Survival curve of WT and *Phlpp1*[-/-] mice i.p. injected with 15 mg/kg LPS. Values are expressed as percent survival of 16 mice for each genotype. \*p<0.05 by log-rank test. (**C–E**) ELISA showing IL-6 (**C**), IL-1β (**D**) and IL-10 (**E**) levels in serum at the indicated times after i.p. injection of 10 mg/kg LPS. Data represent mean ± SEM. Statistics analyzed by Student's *t*-test \*p<0.05, \*\*p<0.01.
DOI: https://doi.org/10.7554/eLife.48609.002

signaling, interferon-β response, and tumor necrosis factor (TNF) signaling-dependent pathways (*Figure 2B*). Genes significantly decreased in *Phlpp1*[-/-] compared to WT macrophages were enriched most significantly in nodes related to central carbon metabolism, and to a lesser extent, chronic inflammatory responses and LPS signaling (*Figure 2B*).

To gain insight into gene regulatory mechanisms affected by loss of PHLPP1, we performed de novo motif analysis of the promoters of upregulated genes in *Phlpp1*[-/-] macrophages using Hypergeometric Optimization of Motif EnRichment (HOMER), a suite of tools for motif discovery and Next Generation Sequencing (NGS) analysis (*Heinz et al., 2010*). This algorithm defines motifs that are statistically enriched in a targeted promoter list compared to random promoter sequences with comparable GC content. The analysis revealed significant enrichment of STAT (p<10$^{-18}$) and IRF (p<10$^{-9}$) motifs (*Figure 2C*) in the promoters of genes whose expression was statistically increased in *Phlpp1*[-/-] macrophages compared to WT macrophages. Of the 199 genes with elevated expression, 46% of the genes had promoters with a consensus STAT binding motif, 51% had promoters with a potential binding site for IRF family of transcription factors, and 26% had promoters with predicted binding sites for both STAT and IRF (*Figure 2D*). We selected for further analysis three genes whose expression was elevated in the *Phlpp1*[-/-] compared to WT macrophages and which had proximal STAT1 binding motifs in their promoters: *Cd69*, *Ifit2*, and *Gbp5*. Normalized mRNA-Seq data for each of these three genes confirmed elevated mRNA levels in *Phlpp1*[-/-] macrophages compared to WT macrophages (*Figure 2E–G*). Thus, loss of PHLPP1 leads to sustained KLA-induced expression of genes involved in inflammation, of which 46% have predicted STAT motifs in their proximal regulatory regions.

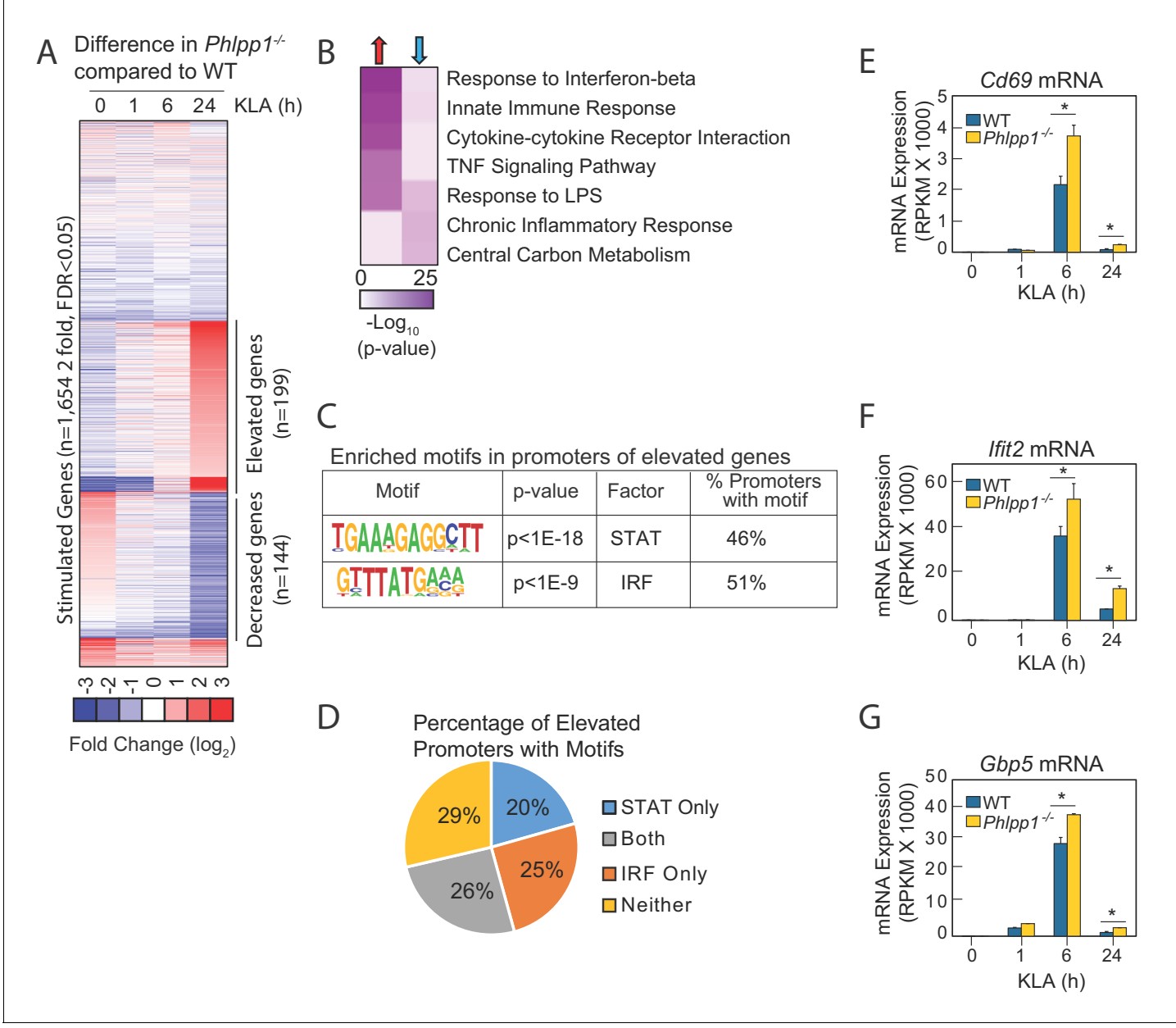

**Figure 2.** Loss of PHLPP1 modulates the expression of inflammatory genes in macrophages. (A) Heat map for mRNA-Seq expression of the 1,654 mRNA transcripts regulated greater than two-fold with a FDR < 0.05 in BMDMs isolated from WT or *Phlpp1*$^{-/-}$ animals treated with 100 ng/ml KLA for 1, 6 or 24 hr. Data represent the log$_2$ difference between the mRNA expression in *Phlpp1*$^{-/-}$ macrophages compared to wild-type macrophages. (B) Gene ontology analysis for the 199 elevated (red arrow) or 144 decreased (blue arrow) transcripts in *Phlpp1*$^{-/-}$ macrophages compared to wild-type macrophages. (C) De novo motif analysis using HOMER (Hypergeometric Optimization of Motif EnRichment) for the 199 promoters corresponding to the genes elevated in the *Phlpp1*$^{-/-}$ macrophages. (D) Pie graph showing the percentage of promoters of elevated genes that contain STAT or IRF binding motifs. (E–G) Normalized mRNA-Seq values for (E) *Cd69* (F) *Ifit2* and (G) *Gbp5* mRNA in BMDMs isolated from WT or *Phlpp1*$^{-/-}$ animals treated with 100 ng/ml KLA for 0, 1, 6, or 24 hr. RPKM – Reads Per Kilobase Million. Values are expressed as mean ± SEM. *p<0.05 (Student's *t*-test) compared to WT cells. See also ***Supplementary file 1*** and ***Supplementary file 2***.

DOI: https://doi.org/10.7554/eLife.48609.003

If PHLPP1 suppresses STAT-regulated gene transcription, we reasoned that 1] knockdown of a STAT family member should reduce transcription of the same genes affected by loss of PHLPP1 and 2] knockdown of PHLPP1 should enhance STAT binding to its promoters. STAT1 is required for LPS-induced gene expression in macrophages (***Ohmori and Hamilton, 2001***) and implicated as a

PHLPP1 target in iNOS regulation (*Alamuru et al., 2014*). STAT1 knockdown by siRNA in thioglycollate-elicited peritoneal macrophages resulted in a 2-fold reduction in KLA-induced transcription of *Cd69*, *Ifit2*, and *Gbp5* at 6 hr compared to a control siRNA transfection, with transcript levels dropping to near baseline by 24 hr (*Figure 3A–C*). The effect of PHLPP1 knockdown on STAT1 promoter occupancy was examined by chromatin immunoprecipitation (ChIP) using STAT1-specific antibodies. KLA induced STAT1 binding to the promoters of *Cd*, *Ifit2*, and *Gbp5*, with maximal binding observed 1 hr post stimulation, followed by a decay in binding to near baseline after 24 hr (*Figure 3D–F*). In contrast, binding to these promoters was enhanced and sustained in *Phlpp1*[-/-] macrophages relative to WT cells. The degree of enhancement and the kinetics of activation/resolution varied depending on the gene examined: PHLPP1 loss had the most robust early effect (1 hr) on the *Ifit2* promoter and at a later time (24 hr) on the *Cd69* promoter. Thus, PHLPP1 suppresses KLA-stimulated binding of STAT1 to its promoters and thereby reduces transcription of its target genes.

## PHLPP1 binds to STAT1 and dephosphorylates Ser727

We next examined whether PHLPP1 affects the phosphorylation state of the two regulatory STAT1 phosphorylation sites, Ser727 and Tyr701. Primary BMDMs were isolated from WT and *Phlpp1*[-/-] mice and the kinetics and magnitude of KLA-triggered phosphorylation at each of the two STAT1 sites were compared. Loss of PHLPP1 in BMDMs led to a robust increase in STAT1 phosphorylation on the regulatory site Ser727 but did not affect Tyr701 phosphorylation (*Figure 4A–B*). PHLPP1 loss also resulted in an increase in Erk phosphorylation at its activation loop sites, as previously reported (*Reyes et al., 2014*). Incubation of *in vitro* phosphorylated STAT1 with immunoprecipitated FLAG-tagged PHLPP1 resulted in dephosphorylation at Ser727, suggesting that PHLPP1 directly dephosphorylates STAT1 (*Figure 4C*). Furthermore, overexpression of PHLPP1 in HEK-293T cells reduced IFNγ-dependent phosphorylation of STAT1 on Ser727 but not on Tyr701 (*Figure 4D–E*). Thus, PHLPP1 selectively dephosphorylates the Ser727 regulatory phosphorylation on STAT1 *in vitro* and in cells.

Because the abundance of PHLPP1 in the cell is much lower than other phosphatases such as PP2A (*Hein et al., 2015*), we next sought to determine whether regulation of STAT1 promoter activity was solely due to PHLPP1 phosphatase activity or occurred in combination with other phosphatases. Taking advantage of the insensitivity of PHLPP phosphatases to the PP1/PP2A inhibitor okadaic acid (OA) (*Gao et al., 2005*), we examined whether OA treatment affected KLA-dependent changes on Ser727 phosphorylation in primary BMDMs from WT mice. *Figure 5A–B* shows that the KLA-induced increase in Ser727 phosphorylation was relatively insensitive to OA, under conditions where the phosphorylation of Erk (at Thr202/Tyr204) and Akt (at Thr308) was significantly increased upon OA addition. These data are consistent with PHLPP1, a PP2C family member, being the primary regulator of phosphorylation on the activity-tuning Ser727 site of STAT1.

We next addressed whether enhanced promoter binding of STAT1 upon loss of PHLPP1 resulted in enhanced transcriptional activation using a luciferase reporter assay. WT or *Phlpp1*[-/-] mouse embryonic fibroblasts (MEFs) were co-transfected with a firefly luciferase reporter construct containing GAS promoter elements, as well as a renilla luciferase controlled by a constitutive CMV promoter as an internal control. STAT1 promoter activity was assessed by monitoring luminescence following IFNγ stimulation. *Figure 6A* shows that STAT1 promoter activity was significantly higher in *Phlpp1*[-/-] MEFs compared to WT MEFs at both 6 hr and 24 hr. Pre-treatment of cells with okadaic acid, under conditions that increased the phosphorylation of PP2A-sensitive substrates (see *Figure 5A* and *Figure 6—figure supplement 1*), had no effect on STAT1 promoter activity (*Figure 6A*); note that treatment with a PKC inhibitor also had no effect on STAT1 promoter activity (*Figure 6—figure supplement 2*). Because STAT1 functions in the nucleus, we next asked whether PHLPP1 regulation of STAT1 occurs in the cytoplasm or nucleus. To this end, we assessed the effect of expressing either the PP2C domain of PHLPP1 or a nuclear-targeted (NLS) PP2C domain of PHLPP1 (*Figure 6B*) on IFNγ-induced STAT1 promoter activity via the GAS luciferase assay. The overexpressed PP2C domain of PHLPP1 (*Figure 6C*, blue) was considerably less effective in inhibiting STAT1 promoter activity compared to full-length PHLPP1 (*Figure 6C*, red). However, forcing the PP2C domain into the nucleus by attaching an NLS to its N-terminus inhibited STAT1 promoter activity as effectively as overexpression of full-length PHLPP1 (*Figure 6C*, orange). Analysis of the subcellular localization of the constructs used in this experiment revealed that full-length PHLPP1 was primarily cytosolic, the isolated PP2C domain had increased nuclear localization, and the NLS-PP2C was enriched in the

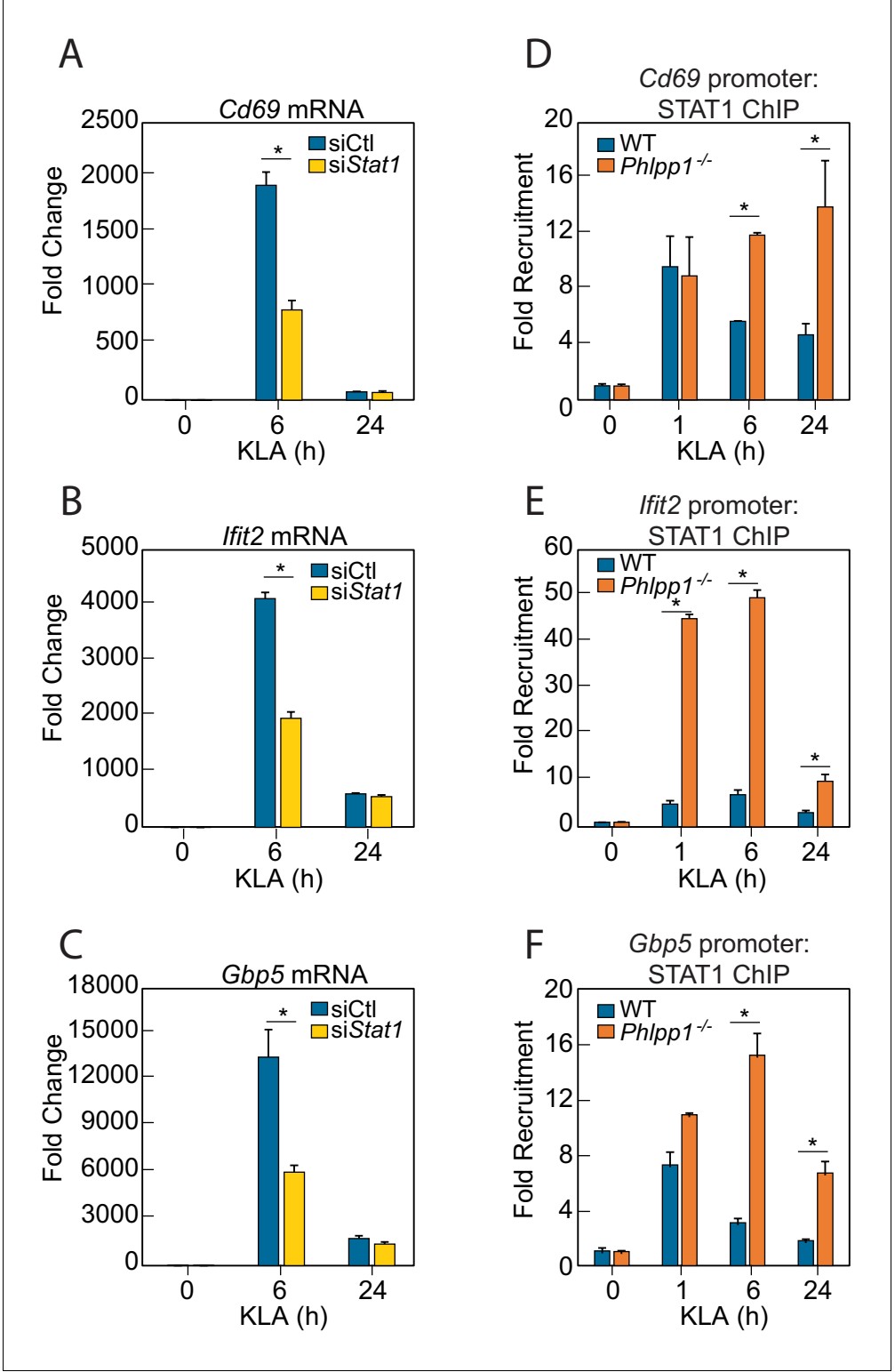

**Figure 3.** PHLPP1 controls STAT1 genomic recruitment and STAT1-dependent gene expression. (A–C) Quantitative PCR analysis for (A) *Cd69* (B) *Ifit2* and (C) *Gbp5* mRNA isolated from thioglycollate-elicited peritoneal macrophages treated with control siRNA (siCtl) or siRNA specifically targeting *Stat1* (si*Stat1*) and subsequently treated with vehicle or 100 ng/ml KLA for 6 or 24 hr. Values are expressed as mean ± SEM from replicate experiments. *p<0.05 (Student's *t*-test) compared to siCtl treated cells. (D–F) Quantitative PCR analysis of ChIPs for STAT1 at the (D) *Cd69* (E) *Ifit2* and (F) *Gbp5* promoter in BMDMs isolated from WT or *Phlpp1*⁻/⁻ animals and

*Figure 3 continued on next page*

*Figure 3 continued*

treated with 100 ng/ml KLA for 0, 1, 6 or 24 hr. Values are expressed as mean ± SEM. *p<0.05 (Student's *t*-test) compared to WT cells.

DOI: https://doi.org/10.7554/eLife.48609.004

nucleus (*Figure 6—figure supplement 3*). To address whether PHLPP1 catalytic activity is required for STAT1 regulation, we utilized a phosphatase-dead PP2C domain in which two active site residues, Asp1210 and Asp1413 (*Sierecki and Newton, 2014*) were mutated to Ala (DDAA). The catalytically-inactive NLS-PP2C was no longer able to suppress STAT1 activity (*Figure 6C*, purple); immunofluorescence confirmed its nuclear localization (*Figure 6—figure supplement 3*). Thus, both the catalytic activity and nuclear localization of PHLPP1 are necessary for it to regulate STAT1 transcriptional activity.

## PHLPP1 has a bipartite nuclear localization signal in its N-Terminal extension

Bioinformatics analysis of the sequence of PHLPP1 using SeqNLS (*Lin and Hu, 2013*) revealed a potential Arg-rich bipartite NLS ($^{92}$RRRRR-X-$^{122}$RRGRLKR) in the N-terminal extension unique to the

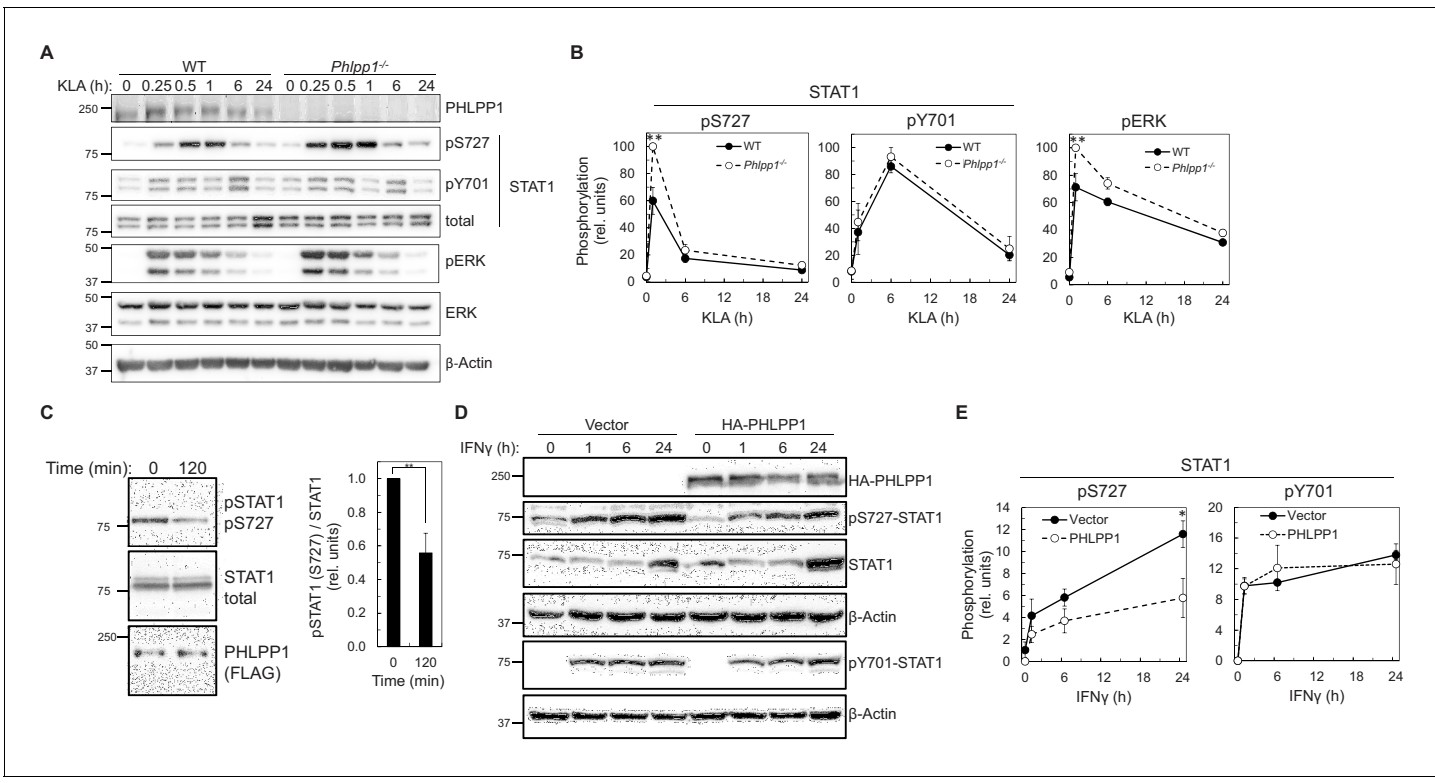

**Figure 4.** PHLPP1 regulates STAT1 phosphorylation on Ser727. (**A**) Western blot analysis of primary BMDM from WT or *Phlpp1$^{-/-}$* mice treated with 100 ng/ml KLA for the indicated times and probed with the indicated antibodies. (**B**) Ratio of pSTAT1 (S727):total STAT1, pSTAT1 (Y701):total STAT1 or phosphoERK (T202/Y204):total ERK normalized to the highest value; data represent the mean ± SEM of five independent experiments as in (**A**). **p<0.01 (Student's *t*-test) compared to WT cells. (**C**) Western blot analysis of an *in vitro* phosphatase assay of purified and phosphorylated STAT1 and immunoprecipitated FLAG-PHLPP1, incubated for 0 or 120 min at 30℃ (on the left). On the right, quantification of pSTAT1 (S727) divided by total STAT1 and normalized to 0 time point. Values are expressed as mean ± SEM of three independent experiments. **p<0.01 (Student's *t*-test). (**D**) Western blot analysis of HEK-293T cells over-expressing vector control (Vector) or HA-tagged PHLPP1 and treated with 10 ng/ml IFNγ for 0, 1, 6, or 24 hr. (**E**) Graphs represent the quantification of three independent experiments as presented in (**D**). Values are expressed as mean relative units of pSTAT1 (S727) or (Y701) divided by β-Actin and normalized to vector 0 hr ± SEM. *p<0.05 (Student's *t*-test) compared to vector control expressing cells. See also *Figure 6—figure supplement 1*.

DOI: https://doi.org/10.7554/eLife.48609.005

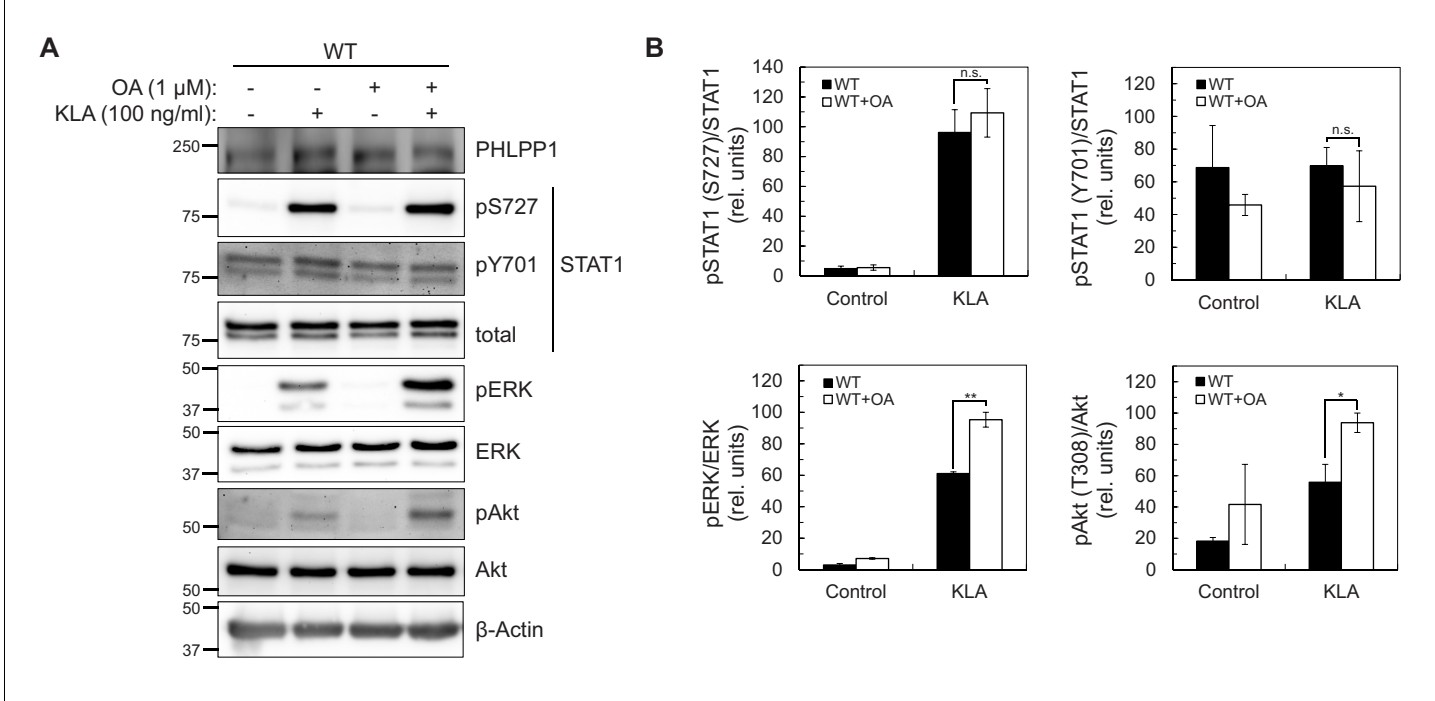

**Figure 5.** STAT1 Ser727 phosphorylation and transcriptional activity are insensitive to okadaic acid. (**A**) Western blot analysis of primary BMDMs from WT or *Phlpp1*[-/-] mice treated with 100 ng/ml KLA for 0 or 30 min followed by treatment with 1 µM OA or DMSO control for an additional 15 min and probed with the indicated antibodies; pAkt antibody recognizes phosphorylated Thr308. (**B**) Graphs represent the quantification of three independent experiments as presented in (**A**). Values are expressed as the mean ± SEM of the ratio of pSTAT1 (S727) to total STAT1 normalized to the highest value; *p<0.05, **p<0.01 and n.s.- non-significant (Student's *t*-test).

DOI: https://doi.org/10.7554/eLife.48609.006

PHLPP1 isozyme (*Figure 6D*). To test whether these basic segments function as an NLS, we examined the subcellular localization in HeLa cells of the NTE alone or NTE in which the basic residues in each or both halves of the potential bipartite NLS were mutated to Ala (*Figure 6E*). Immunofluorescence revealed that the NTE localizes to the nucleus. Mutation of the first NLS or the second NLS increased the distribution of the NTE to the cytosol, and mutation of both decreased the nuclear: cytoplasmic ratio to be comparable to that of a construct of the NTE with a strong Nuclear Export Signal (NES) (LALKLAGLDI from PKI; see *Wen et al., 1995a*) (*Figure 6F*). Full-length PHLPP1 was primarily cytosolic, leading us to ask whether there may also be an NES. Bioinformatics analysis of the primary sequence identified a potential Leu-rich NES (*Xu et al., 2015*) immediately following the last LRR and preceding the phosphatase domain (*Figure 6—figure supplement 4*). Attachment of this 14-residue sequence to the N-terminus of the NTE resulted in distribution of the NTE to the cytosol (*Figure 6—figure supplement 4*). Thus, PHLPP1 nuclear localization is controlled by a bipartite NLS in the NTE and is opposed by an NES following the LRR. Lastly, we examined the effect of mutating the NLS on the ability of full-length PHLPP1 to reduce STAT1 transcriptional activity as assessed using the GAS promoter assay. The reduction in IFNγ-induced STAT1 activity resulting from PHLPP1 overexpression (*Figure 6G*, red) was abolished upon mutation of NLS2 (*Figure 6G*, brown) or both halves of the NLS (NLS1/2) (*Figure 6G*, purple). Mutation of NLS1 had an intermediate effect (*Figure 6G*, blue). These data reveal that a bipartite NLS in the NTE of PHLPP1 localizes PHLPP1 to the nucleus, where it suppresses the transcriptional activity of STAT1.

We next assessed which domain of PHLPP1 contributes to the observed regulation of STAT1 activity on the GAS promoter. Overexpression of full-length PHLPP1 in HEK-293T cells markedly reduced GAS promoter activity (*Figure 7A*, red) compared to the vector only control (*Figure 7A*, black). A construct of PHLPP1 lacking the NTE (deletion of first 512 amino acids of its N-terminus; PHLPP1ΔNTE, blue) was less effective than full-length PHLPP1 in reducing STAT1 activity, whereas a construct comprised of just the NTE (amino acids 1–512, green) caused a significant increase in GAS

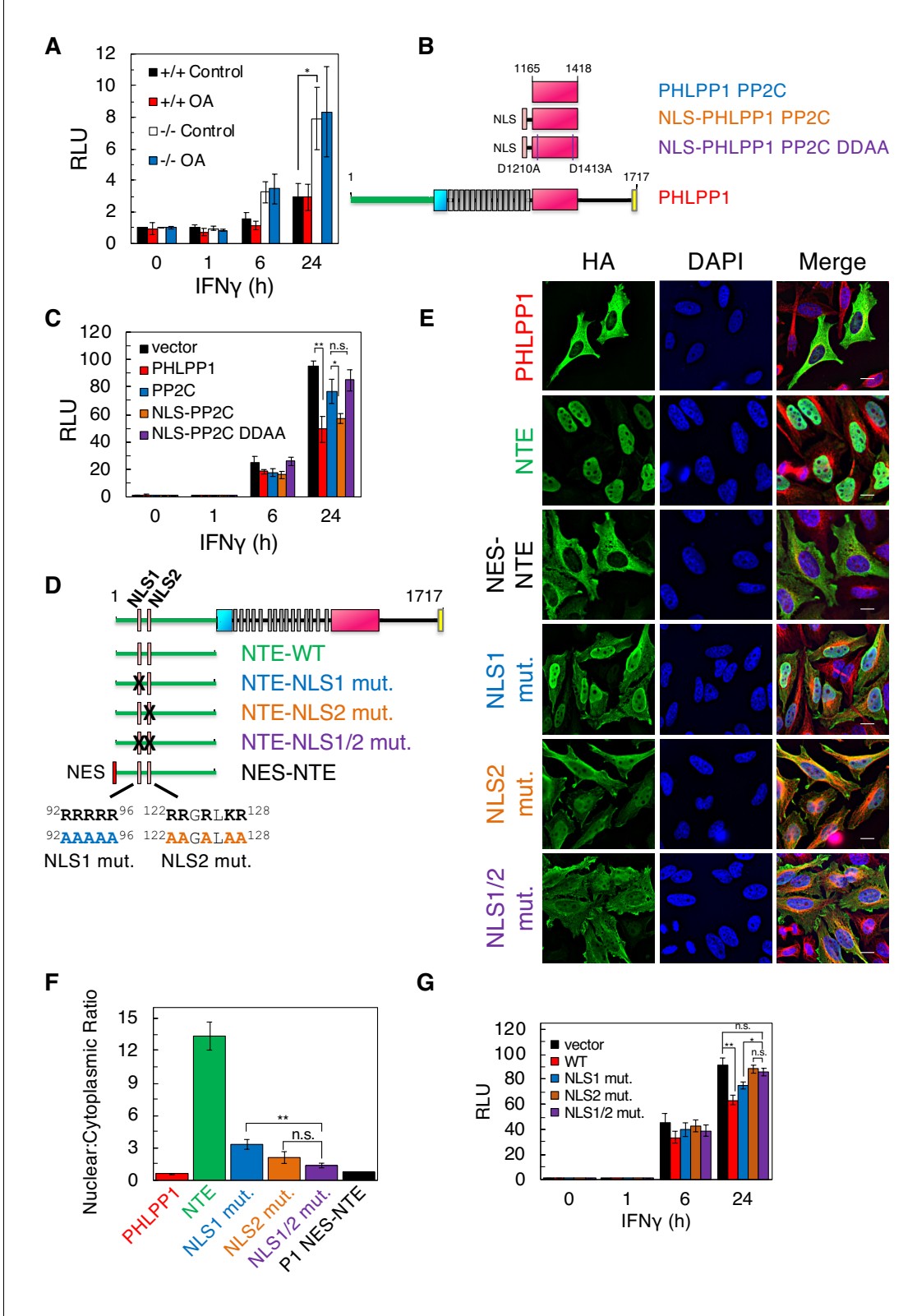

**Figure 6.** PHLPP1 suppresses STAT1 transcriptional activity by a mechanism that depends on its catalytic activity and an NLS in its N-Terminal Extension. (A) Luciferase reporter assay in WT (+/+) and *Phlpp1*[-/-] (-/-) MEFs over-expressing GAS luciferase reporter and treated with 10 ng/ml IFNγ for 0, 1, 6, or 24 hr in combination with 1 μM OA or DMSO control treatment for 15 min. Values are expressed as mean of relative light units (RLU) ± SEM of three independent experiments. *p<0.05 (Student's *t*-test). See also *Figure 6—figure supplement 1*. (B) Schematic of HA-tagged PHLPP1 constructs

*Figure 6 continued on next page*

Figure 6 continued

used in this study: the PP2C domain of PHLPP1 (PP2C), nuclear targeted PP2C with NLS (NLS- PP2C), NLS-PP2C with active site residues Asp1210 and Asp1413 mutated to Ala (NLS-PP2C DDAA), and full-length PHLPP1 (PHLPP1). (**C**) Luciferase reporter assay in HEK-293T cells over-expressing GAS luciferase reporter in combination with either vector control (vector, black) or the constructs described in (**B**) and treated with 10 ng/ml IFNγ for 0, 1, 6, or 24 hr. Values are expressed as mean RLU ± SEM of four independent experiments. All data points at 24 hr were significant against each other except for vector to PP2C, vector to NLS-PP2C DDAA, P1 to NLS-PP2C, and PP2C to NLS-PP2C DDAA. *p<0.05, **p<0.01 (Student's *t*-test). (**D**) Schematic showing position and sequence of bipartite NLS in the NTE, and NLS mutants used in this study. (**E**) HeLa cells over-expressing the constructs used in *Figure 6D* were stained for HA (green), α-Tubulin (red), and DAPI (blue). Scale bar indicates 15 μm. (**F**) The Nuclear to Cytoplasmic ratio was calculated for each construct (300 cells per construct) and values are expressed as mean ± SEM. All data points were significant against each other except for NLS1 to NLS2, and NLS2 to NLS1/2. **p<0.01, n.s. – non-significant (Student's *t*-test). (**G**) Luciferase reporter assay in HEK-293T cells over-expressing a GAS luciferase reporter in combination with either vector control (vector, black) or the constructs described in (**D**) however in the context of a full-length PHLPP1 and treated with 10 ng/ml IFNγ for 0, 1, 6, or 24 hr. Values are expressed as mean RLU ± SEM of eight independent experiments. *p<0.05, **p<0.01, n.s. - non-significant (Student's *t*-test).

DOI: https://doi.org/10.7554/eLife.48609.007

The following figure supplements are available for figure 6:

**Figure supplement 1.** STAT1 phosphorylation and transcriptional activity are insensitive to okadaic acid.

DOI: https://doi.org/10.7554/eLife.48609.008

**Figure supplement 2.** Luciferase reporter assay in HEK-293T cells over-expressing GAS luciferase reporter and treated with 10 ng/ml IFNγ for 0, 1, 6, or 24 hr followed by 250 nM Gö6983 for 10 min, 1 μM staurosporine for 30 min, or DMSO control.

DOI: https://doi.org/10.7554/eLife.48609.009

**Figure supplement 3.** The phosphatase activity of PHLPP1 is important for the regulation of STAT1 activity.

DOI: https://doi.org/10.7554/eLife.48609.010

**Figure supplement 4.** PHLPP1 has an NES.

DOI: https://doi.org/10.7554/eLife.48609.011

promoter activity, suggesting a dominant-negative function of this segment. Co-immunoprecipitation assays revealed a robust interaction of STAT1 with the immunoprecipitated NTE of PHLPP1, in contrast to barely detectable binding of STAT1 to PHLPP1 lacking the NTE (*Figure 7B*). Intermediate binding was observed between STAT1 and full-length PHLPP1. Quantification of three independent experiments revealed that the isolated NTE of PHLPP1 binds STAT1 approximately five times more

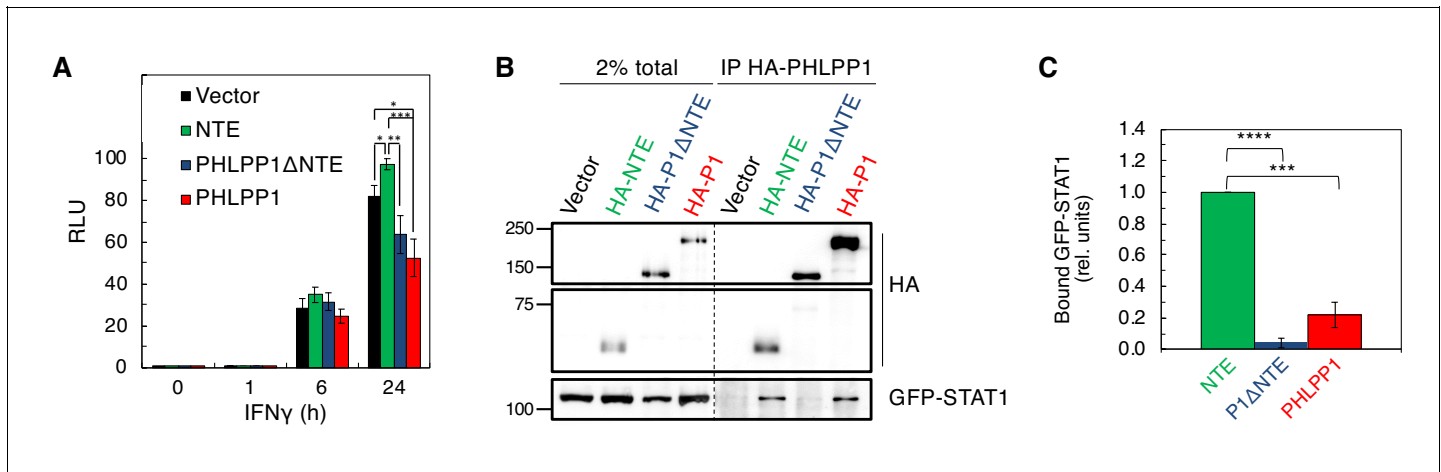

**Figure 7.** STAT1 associates with the N-Terminal Extension of PHLPP1. (**A**) Luciferase reporter assay in HEK-293T cells over-expressing GAS luciferase reporter in combination with either vector control (Vector, black), PHLPP1 NTE (NTE, green), PHLPP1ΔNTE (blue), or PHLPP1 (red) and treated with 10 ng/ml IFNγ for 0, 1, 6, or 24 hr. Values are expressed as mean of RLU ± SEM of five independent experiments. All data points at 24 hr were significant against each other except for vector to PHLPP1ΔNTE, and PHLPP1ΔNTE to PHLPP1. *p<0.05, **p<0.01, ***p<0.001 (Student's *t*-test). (**B**) Western blot analysis of detergent-solubilized lysate of HEK-293T cells transfected with vector control (Vector), HA-tagged NTE of PHLPP1 (HA-NTE), PHLPP1 with the NTE deleted (HA-P1ΔNTE) or full-length PHLPP1 (HA-P1) and immunoprecipitated (IP) using HA antibody; blots were probed for co-IP of STAT1 tag using GFP antibody. (**C**) Quantification of GFP-STAT1 IP divided by HA IP and normalized to HA-NTE IP. Values are expressed as mean ± SEM of three independent experiments. ***p<0.001, ****p<0.0001 (Student's *t*-test).

DOI: https://doi.org/10.7554/eLife.48609.012

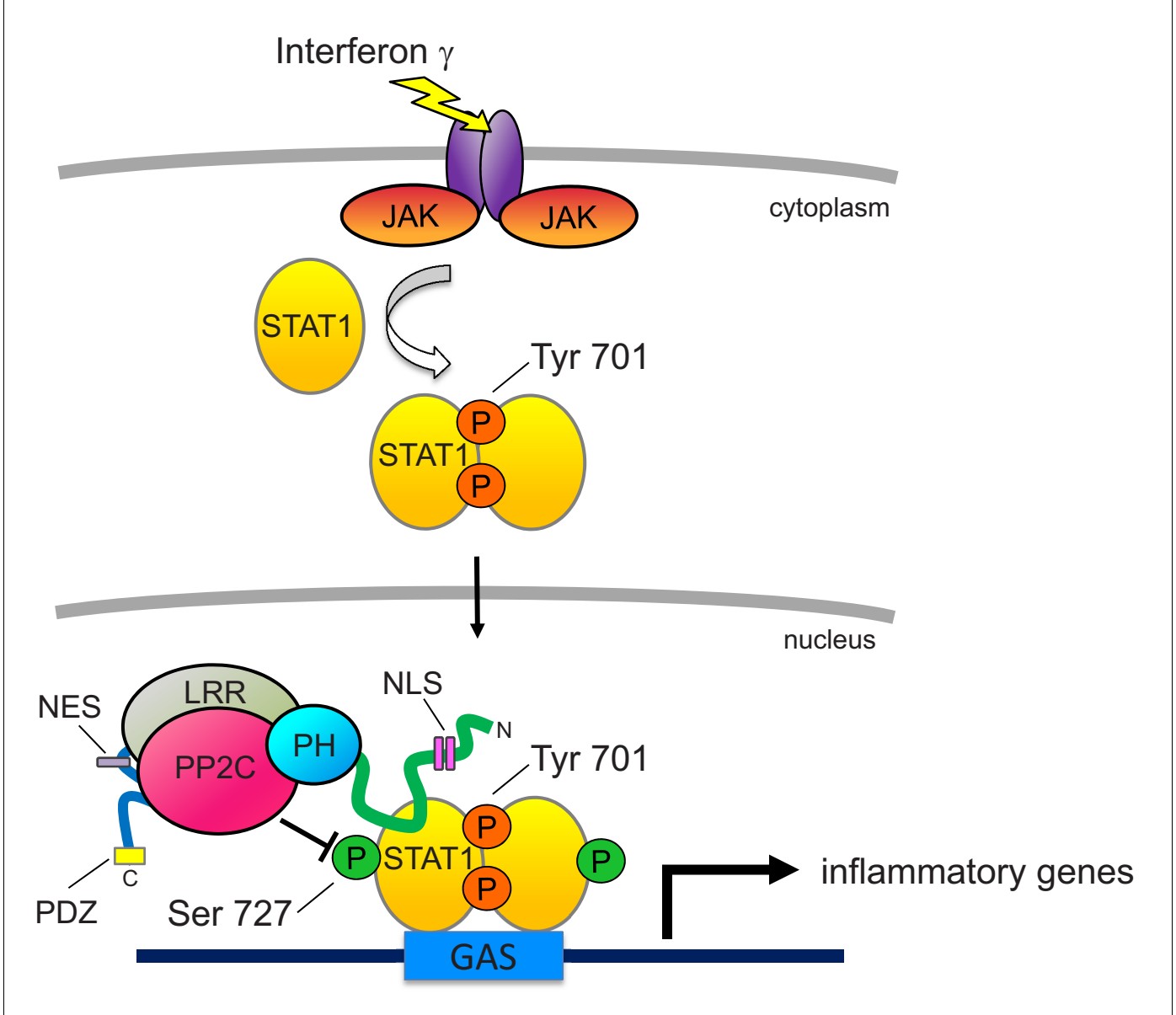

**Figure 8.** Proposed model for PHLPP1-dependent suppression of STAT1 activity. Binding of IFNγ to IFN receptors results in their dimerization and phosphorylation, promoting the recruitment of JAK, which phosphorylates STAT1 on Tyr701. This promotes the dimerization of STAT1 and its translocation into the nucleus where it binds the GAS promoter to allow the transcription of inflammatory response genes. Activity of STAT1 is enhanced by phosphorylation on Ser727. However, nuclear-localized PHLPP1, which binds STAT1 via its N-terminal extension, tunes the activity of STAT1 by directly dephosphorylating this site to keep activity finely controlled. Loss of PHLPP1 results in poor resolution of inflammatory response. The nuclear localization of PHLPP1 is controlled by a bipartite NLS (pink rectangles) in the NTE (green line) and an NES (purple rectangle) in the segment between the LRR (gray oval) and the PP2C domain (red oval).

DOI: https://doi.org/10.7554/eLife.48609.013

strongly than full-length PHLPP1 and 26 times more strongly than PHLPP1 lacking the NTE (*Figure 7C*). These data reveal that the NTE of PHLPP1 interacts with STAT1 and reduces its promoter activity.

## Discussion

The finding that *Phlpp1*[−/−] mice are protected from LPS-induced death allowed us to identify PHLPP1 as a physiologically relevant phosphatase in the overall innate immune response. It is likely that this immunoregulatory phenotype reflects roles of PHLPP1 in several immune cell types, and future studies of mice with cell-specific deletions of *Phlpp1* will be of great interest. Investigation of *Phlpp1*[−/−] macrophages indicates a significant role in counter-regulation of STAT1-dependent transcription that emerges as a secondary response to TLR4 ligation. Our mechanistic analyses show that PHLPP1 dephosphorylates STAT1 on a key regulatory site to suppress its transcriptional activity towards an array of genes involved in mounting an inflammatory response to IFNγ. Specifically, PHLPP1 directly dephosphorylates Ser727 on STAT1 *in vitro* and specifically suppresses phosphorylation of Ser727, but not Tyr701, on STAT1 in cells, correlating to decreased transcriptional activity of STAT1 at one of its major binding sites, the GAS promoter. The intrinsic catalytic activity and nuclear localization of PHLPP1 is required for this transcriptional regulation; while the isolated PP2C domain is not efficient in suppressing GAS promoter activity, forcing the PP2C domain into the nucleus is as effective as the full-length phosphatase in controlling transcriptional activity. Nuclear localization of the full-length enzyme is driven by a bipartite NLS we identify in the NTE. Elimination of PHLPP1 results in global changes in KLA-dependent transcriptional regulation, with 20% of the approximately 2000 genes whose expression changes upon KLA stimulation differing by more than two-fold in BMDMs from *Phlpp1*[−/−] mice compared to WT mice.

Phosphorylation of STAT1 on Ser727 has been proposed to occur following the binding of the Tyr-phosphorylated STAT1 dimer to chromatin (*Sadzak et al., 2008*). Ser727 phosphorylation on the C-terminal transactivation domain of STAT1 is necessary for maximal transcriptional activity. Identification of PHLPP1 as a phosphatase that opposes this phosphorylation provides a mechanism to counter-regulate the activity of this key transcription factor. Several lines of evidence suggest that PHLPP1 may be the major phosphatase that controls this regulatory site. First, genetic depletion of PHLPP1 increases both STAT1 Ser727 phosphorylation and transcriptional activity at the GAS promoter, whereas PHLPP1 overexpression decreases both STAT1 Ser727 phosphorylation and transcriptional activity at the promoter. Second, both the IFNγ-induced phosphorylation of Ser727 and resulting increase in transcriptional activity at the GAS promoter are insensitive to OA, a phosphatase inhibitor that is ineffective towards PP2C family members but highly effective towards the abundant PP2A in cells. The insensitivity of STAT1 Ser727 phosphorylation to OA is consistent with PHLPP1 directly dephosphorylating this site in cells, a reaction it catalyzes *in vitro*. Furthermore, although PHLPP1 does suppress the signaling output of Akt (by dephosphorylating Ser473; see *Gao et al., 2005*) and Erk (by reducing the steady-state levels of RTKs; see *Reyes et al., 2014*), its effect on STAT1 is unlikely to involve either of these targets because the activities of both kinases are sensitive to OA. Nor are the effects on Ser727 a result of PHLPP1 reducing PKC steady-state levels (*Baffi et al., 2019*; *Gao et al., 2008*), as the general PKC inhibitor Gö6983 did not alter GAS promoter activity (*Figure 6—figure supplement 2*). Third, genetic depletion of either PHLPP1 or STAT1 has opposing effects on transcriptional targets of STAT1: whereas KLA causes a larger increase in mRNA of *Cd69*, *Ifit2*, and *Gbp5* in BMDMs from *Phlpp1*[−/−] mice compared to WT mice, a reduction in these transcripts is observed upon STAT1 knockdown. Lastly, we have previously shown that PHLPP1 regulates transcription of genes and binds chromatin (*Reyes et al., 2014*). Cumulatively, these data are consistent with PHLPP1 being the major phosphatase to oppose the activating phosphorylation of STAT1 on Ser727, thereby limiting its transcriptional activity.

The interaction of PHLPP1 with STAT1, mediated by its NTE, affords fidelity and specificity in its dephosphorylation of the transcription factor. PHLPP1 binding to STAT1 is consistent with this multivalent protein utilizing its protein-interaction domains to position it near its substrates, either via direct interaction or by binding protein scaffolds, such as PDZ domain proteins that coordinate Akt signaling (*Li et al., 2011*). Such coordination is essential for its dephosphorylation of relevant substrates, in part due to the low catalytic activity of the phosphatase (approximately one reaction per sec towards peptide substrates, over an order of magnitude lower than that of the related phosphatase PP2Cα; see *Sierecki and Newton, 2014*). The importance of enzyme proximity to its substrate is best illustrated with Akt, where deletion of the last three amino acids of PHLPP1 to remove the PDZ ligand abolishes the ability of PHLPP1 to dephosphorylate Akt in cells (*Gao et al., 2005*). Thus,

binding of PHLPP1 via its NTE to STAT1 affords an efficient mechanism to restrict its activity by directly opposing its phosphorylation in the nucleus (see *Figure 8*).

The regulation of STAT1 by PHLPP1 occurs in the nucleus, and we identify motifs in the phosphatase that control both the entry into (NLS) and exit from (NES) the nucleus. First, we identify a bipartite NLS in the NTE of PHLPP1 whose integrity is necessary for the phosphatase to regulate the transcriptional activity of STAT1. Second, we identify an NES in the segment between the LRR and PP2C domain that drives export out of the nucleus. Under the 'unstimulated' conditions of our immunofluorescence, PHLPP1 localized primarily to the cytosol, suggesting masking of the NLS and exposure of the NES. Inputs that regulate the exposure of the NLS and NES are likely important regulators of PHLPP1 function.

Our transcriptomic data support a key role for PHLPP1 in the resolution of the inflammatory response specific to genes downstream of type II IFN signaling pathways. This suggests the possibility that PHLPP1 can selectively discriminate between inflammatory promoters that are differentially regulated by distinct transcription factor families. Surprisingly, over 50% of the inflammatory genes that fail to properly resolve in the macrophages from *Phlpp1*[−/−] mice contain a consensus STAT-binding motif in their proximal promoters. Our studies have demonstrated a direct interaction between PHLPP1 and STAT1, thus it is highly likely that PHLPP1 is recruited to gene promoters through its association with STAT1. Elevated STAT1 occupancy and delayed dismissal kinetics of STAT1 from its target promoters in *Phlpp1*[−/−] macrophages indicate a major function of PHLPP1-dependent dephosphorylation in termination of STAT1 signaling and its dismissal from chromatin.

Germline mutations that impair STAT1 function, by reducing either Tyr701 phosphorylation (L706S) or DNA binding (Q463H and E320Q), increase the susceptibility of otherwise healthy patients to mycobacterial and viral infection (*Chapgier et al., 2006*; *Dupuis et al., 2001*). This increased susceptibility was proposed to arise because of reduced transcription of genes involved in bacterial and viral immunity from the GAS and ISRE promoters, respectively. Similarly, genetic ablation of *Stat1* on the background of a mouse that has enhanced TLR4 signaling (because of deletion of *Il6st*, a key regulator of systemic inflammatory responses during LPS-mediated endotoxemia) provides protection against LPS-induced toxemic death compared to mice with normal STAT1 levels (*Luu et al., 2014*). Although the current study does not provide direct evidence that enhanced phosphorylation of STAT1 causes the protective effect of PHLPP1 loss on both *E. coli*-induced sepsis and LPS-induced endotoxemia in mice, our data indicate that PHLPP1 inhibitors could be explored as adjunctive therapies to antibiotics and supportive care of patients with Gram-negative sepsis, a leading cause of mortality in intensive care units.

## Materials and methods

### Key resources table

| Reagent type (species) or resource | Designation | Source or reference | Identifiers | Additional information |
|---|---|---|---|---|
| Cell line (*Mus musculus*) | Primary Bone Marrow Derived Macrophage Cells from *Phlpp1*[+/+] mice | This paper | WT BMDM | male, 6–8 week old C57BL/6 mice, Charles River Laboratories |
| Cell line (*Mus musculus*) | Primary Bone Marrow Derived Macrophage Cells from *Phlpp1*[−/−] mice | This paper | *Phlpp1*[−/−] BMDM | male, 6–8 week old C57BL/6 mice, Charles River Laboratories |
| Cell line (*Mus musculus*) | Immortalized MEFs from *Phlpp1*[+/+] mice | Lloyd Trotman Lab, CSHL, PMID: 21840483 | | stably expressing shp53 and GFP |
| Cell line (*Mus musculus*) | Immortalized MEFs from *Phlpp1*[−/−] mice | Lloyd Trotman Lab, CSHL, PMID: 21840483 | | stably expressing shp53 and GFP |
| Cell line (*Homo sapiens*) | HeLa | ATCC | CCL-2 | |
| Cell line (*Homo sapiens*) | HEK-293T | ATCC | CRL-11268 | |
| Cell line (*Mus musculus*) | NCTC clone L929 | ATCC | CCL-1 | L cell, L-929, derivative of Strain L |

*Continued on next page*

*Continued*

| Reagent type (species) or resource | Designation | Source or reference | Identifiers | Additional information |
|---|---|---|---|---|
| Strain, strain background (*Escherichia coli*) | *E. coli* K1 strain RS218 | PMID: 6995336 | | Victor Nizet Lab, University of California San Diego |
| Strain, strain background (*Mus musculus*) | WT and littermate control *Phlpp1⁻ᐟ⁻* mice | PMID: 20080691 | | female, 10 to 14 week old, C57BL/6, for mouse infection studies |
| Antibody | anti-HA (rat, monoclonal) | Roche | 11867425001 | Western Blot (1:1000) |
| Antibody | anti-GFP (rabbit, polyclonal) | Cell Signaling | 2555 | Western Blot (1:1000) |
| Antibody | anti-STAT1 (rabbit, polyclonal) | Cell Signaling | 9172 | Western Blot (1:1000) |
| Antibody | anti-STAT1 pSer727 (rabbit, polyclonal) | Cell Signaling | 9177 | Western Blot (1:1000) |
| Antibody | anti-STAT1 pTyr701 (rabbit, polyclonal) | Cell Signaling | 7649 | Western Blot (1:1000) |
| Antibody | anti-Erk1/2 pThr202/pTyr204 (rabbit, polyclonal) | Cell Signaling | 9101 | Western Blot (1:1000) |
| Antibody | anti-Erk1/2 (rabbit, polyclonal) | Cell Signaling | 9102 | Western Blot (1:1000) |
| Antibody | anti-Akt pThr308 (rabbit, polyclonal) | Cell Signaling | 9275 | Western Blot (1:1000) |
| Antibody | anti-Akt (rabbit, polyclonal) | AbCam | 126811 | Western Blot (1:1000) |
| Antibody | anti-PHLPP1 (rabbit, polyclonal) | Proteintech | 22789–1-AP | Western Blot (1:1000) |
| Antibody | anti-FLAG (mouse, monoclonal) | Sigma-Aldrich | F3165 | Western Blot (1:1000) |
| Antibody | anti-β-Actin (mouse, monoclonal) | Sigma-Aldrich | A2228 | Western Blot (1:2000) |
| Antibody | anti-α-tubulin (mouse, monoclonal) | Sigma-Aldrich | T6074 | Western Blot (1:1000) |
| Antibody | anti-STAT1 (rabbit, polyclonal) | Santa Cruz Biotechnology | sc-345 | Chromatin Immunoprecipitation |
| Antibody | anti-HA (mouse, monoclonal) | BioLegend | 901503 | Immunoprecipitation |
| Antibody | anti-α-tubulin (rabbit, monoclonal) | Cell Signaling | 2125 | Immunofluorescence (1:200) |
| Antibody | anti-Alexa647 (goat anti-Mouse IgG) | Life Technologies | A21235 | Immunofluorescence (1:500) |
| Antibody | anti-Alexa477 (goat anti-Rabbit IgG) | Life Technologies | A11034 | Immunofluorescence (1:500) |
| Recombinant DNA reagent | pCMV 3XFLAG-PHLPP1 WT | This paper | | residues 1–1717 of PHLPP1 |
| Recombinant DNA reagent | pCDNA3 HA-NLS-PHLPP1 PP2C | This paper | | residues 653–906 of PHLPP1 |
| Recombinant DNA reagent | pCDNA3 HA-NLS-PHLPP1 PP2C D1210/1413A | This paper | DDAA | Aspartate at residues 1210 and 1413 mutated to Alanine |
| Recombinant DNA reagent | pCDNA3 HA PHLPP1 NTE-WT | This paper | | residues 1–512 of PHLPP1 |
| Recombinant DNA reagent | pCDNA3 HA-PHLPP1 NTE-NLS1 mut. | This paper | | residues 1–512 of PHLPP1 |

*Continued on next page*

*Continued*

| Reagent type (species) or resource | Designation | Source or reference | Identifiers | Additional information |
|---|---|---|---|---|
| Recombinant DNA reagent | pCDNA3 HA-PHLPP1 NTE-NLS2 mut. | This paper | | residues 1–512 of PHLPP1 |
| Recombinant DNA reagent | pCDNA3 HA-PHLPP1 NTE-NLS1/2 mut. | This paper | | residues 1–512 of PHLPP1 |
| Recombinant DNA reagent | pCDNA3 HA-NES-PHLPP1 NTE | This paper | | residues 1–512 of PHLPP1 |
| Recombinant DNA reagent | pCDNA3 HA-PHLPP1 ΔNTE | PMID: 15808505 | Addgene: 22404 | residues 513–1717 of PHLPP1 |
| Recombinant DNA reagent | pCDNA3 HA-$^{PHLPP1}$NES-PHLPP1 NTE | This paper | | residues 1–512 of PHLPP1 |
| Sequence-based reagent | *Stat1* SMART siRNA pools | Dharmacon | L-058881 | |
| Sequence-based reagent | *Control* SMART siRNA pools | Dharmacon | D-001810-10-05 | |
| Peptide, recombinant protein | murine IFNγ | PeproTech | 315–05 | 10 ng/mL |
| Peptide, recombinant protein | human IFNγ | PeproTech | 300–02 | 10 ng/mL |
| Peptide, recombinant protein | recombinant STAT1 | Biosource | PHF0011 | 0.3 uM |
| Peptide, recombinant protein | recombinant human Cdk1/CyclinB | Millipore | 14–450 | 0.2 uM |
| Peptide, recombinant protein | *E. coli* O111:B4 LPS | Sigma-Aldrich | L4391 | |
| Chemical compound, drug | Okadaic Acid (OA) | Millipore | 459616 | 1 μM |
| Chemical compound, drug | Gö6983 | Calbiochem | 365251 | 250 nM |
| Chemical compound, drug | Staurosporine | Calbiochem | 569397 | 1 μM |
| Chemical compound, drug | KLA | Avanti Polar Lipids | 699500 | 100 ng/mL |
| Chemical compound, drug | RO-3306 | Enzo | ALX-270–463 | 144 uM |
| Commercial assay or kit | Dual-Glo Luciferase Assay System | Promega | E2940 | |
| Commercial assay or kit | IL-6 ELISA Kit | R and D Systems | DY406 | |
| Commercial assay or kit | IL-1β ELISA Kit | R and D Systems | DY401 | |
| Commercial assay or kit | IL-10 ELISA Kit | R and D Systems | DY417 | |

## Materials and antibodies

OA (459616) was purchased from Millipore. Gö6983 (365251) and staurosporine (569397) were purchased from Calbiochem. Antibody against HA (11867425001) was purchased from Roche; antibodies against GFP (2555), STAT1 (9172), phosphorylated Ser727 on STAT1 (9177), phosphorylated Tyr701 on STAT1 (7649), phosphorylated Thr202/Tyr204 on Erk1/2 (9101), total Erk1/2 (9102), and phosphorylated Thr308 on Akt (9275) were purchased from Cell Signaling. Antibody against total Akt (126811) was obtained from AbCam. Antibodies against PHLPP1 were purchased from Cosmo (KIAA0606) and Proteintech (22789–1-AP); antibodies against FLAG (F3165), β-Actin (A2228), and α-tubulin (T6074) were purchased from Sigma-Aldrich. The pcDNA3 HA-tagged PHLPP1 and PHLPP2 constructs for mammalian cell expression were described previously (*Brognard et al., 2007*;

*Gao et al., 2008*; *Gao et al., 2005*). Full-length PHLPP1 was cloned into pCMV 3xFLAG vector (Sigma-Aldrich, E4401). An NLS was cloned to the N-terminus of the PP2C domain of PHLPP1. A double mutant of NLS-PP2C at residues D1210A and D1413A was cloned by site-directed mutagenesis. The HA-tagged PHLPP1 N-terminal extension (PHLPP1 NTE), residues 1–512, was cloned into pcDNA3 vector (Invitrogen). The NLS1 and NLS2 mutations were cloned by site-directed mutagenesis into HA-PHLPP1 NTE. The NES from PKI (LALKLALDI) was cloned into the N-terminus of HA-PHLPP1 NTE. The PHLPP1 NES (residues 1125–1134, LPPKLQELDL) was subcloned directly downstream of the HA-tag in HA-PHLPP1 NTE, to generate HA-$^{PHLPP1}$NES-NTE.

## Isolation and treatment of macrophages

Primary BMDM cells were isolated from male 6- to 8-week-old C57BL/6 mice (Charles River Laboratories). BMDMs were obtained by PBS flush of femurs and tibias (*Weischenfeldt and Porse, 2008*), red blood cells lysed, and remaining cells plated in RPMI 1640 supplemented with 20% fetal bovine serum (FBS, Gibco, cat. 12657–029), 30% L-cell conditioned medium, 100 U/ml penicillin, 100 µg/ml streptomycin, and 2 mM L-glutamine. Cells were seeded in non-tissue culture treated Optilux Petri dishes (BD Biosciences), incubated at 37°C in a 5% $CO_2$ atmosphere for 7 days, then treated with 100 ng/ml KLA (699500, Avanti Polar Lipids) for noted times. Peritoneal macrophages were collected by flushing mouse peritoneal cavities with PBS following 48 hr post peritoneal injection with 3 ml of thioglycolate (*Ray and Dittel, 2010*).

## Cell culture

MEFs from WT or *Phlpp1$^{-/-}$* mice stably expressing shp53 were a kind gift from Lloyd Trotman (CSHL) and have been described previously (*Chen et al., 2011*); MEFs, HEK-293T, and HeLa (ATCC) cells were grown in Dulbecco's modified Eagle medium (DMEM, 10–013-CV, Corning) supplemented with 10% fetal bovine serum (S11150, Atlanta biologicals) and 1% penicillin/streptomycin (15140–122, Gibco) at 37°C in 5% (vol/vol) $CO_2$. Cells used were periodically tested for *Mycoplasma* contamination using a PCR-based protocol (*Uphoff and Drexler, 2011*) and showed no evidence of contamination.

## mRNA isolation and qPCR analysis

RNA was purified using Direct-zol RNA Miniprep Kits (Zymo Research) from triplicate experiments and quantified using a NanoDrop Spectrophotometer (ThermoFisher Scientific). RNA was either reverse transcribed into cDNA for quantitative real-time PCR using gene-specific primers or used for next-generation library preparation. For cDNA generation, one µg of total mRNA was reverse transcribed using the SuperScript III Reverse Transcriptase (ThermoFisher Scientific). The resulting cDNA (25 ng) was used to perform real-time PCR using SYBR Green Master Mix (ThermoFisher Scientific) and 50 nM mix of forward and reverse primers. The real-time PCR values for individual genes were normalized to the house keeping gene, *36B4*, using the ΔΔCT method (*Livak and Schmittgen, 2001*). The primer sequences used in this study are:

*36B4*_qPCR_F AATCTCCAGAGGCACCATTG
*36B4*_qPCR_R CCGATCTGCAGACACACACT
*Cd69*_qPCR_F CTATCCCTTGGGCTGTGTTAAT
*Cd69*_qPCR_R ACATGGTGGTCAGATGATTCC
*Ifit2*_qPCR_F GAGTTTGAGGACAGGGTGTTTA
*Ifit2*_qPCR_R AGACCTCTGCAGTGCTTTAC
*Gbp5*_qPCR_F GGAAGTGCTGCAGACCTATT
*Gbp5*_qPCR_R GCTCTTTCTTGTTCCGCTTTAC

## Next-generation sequence library preparation and analysis

Libraries were prepared from two biological replicates per condition. RNA-Seq libraries were prepared as previously described (*Kaikkonen et al., 2013*). Sequencing libraries were prepared using magnetic beads similar to described previously using barcoded adapters (NextFlex, Bioo Scientific) (*Garber et al., 2012*). Libraries were sequenced for 36 or 50 cycles on an Illumina Genome Analyzer II or HiSeq 2000, respectively, according to the manufacturer's instructions. mRNA-Seq results were trimmed to remove A-stretches originating from the library preparation. Each sequence tag returned

by the Illumina Pipeline was aligned to the mm10 assembly using ELAND allowing up to two mismatches. Only tags that mapped uniquely to the genome were considered for further analysis. Peak finding, MOTIF discovery, and downstream analysis was performed using HOMER, a software suite created for analysis of high-throughput sequencing data (*Heinz et al., 2010*). Detailed instructions for analysis can be found at http://homer.ucsd.edu/homer/. Data visualization was performed using Microsoft Excel, JavaTreeGraph and software packages available in R.

## RNA interference experiments

SMART siRNA pools for examined genes were purchased from Dharmacon (Control: D-001810-10-05, *Stat1*: L-058881). Thioglycollate-elicited peritoneal macrophages were transfected with 30 nM siRNA for 48 hr using Deliver X (Affymetrix) according to the manufacturer's instructions prior to being stimulated with KLA for designated times.

## Chromatin immunoprecipitation

ChIP assays were performed as described before (*Stender et al., 2017*). Cells were crosslinked with 2 mM disuccinimidyl glutarate for 30 min prior to 10 min treatment with 1% formaldehyde. The antibodies used in these studies were: STAT1 (sc-345, Santa Cruz Biotechnology). For the precipitations protein A Dynabeads (10003D, Invitrogen) were coated with antibody prior to pulldown and excess antibody was washed away. Pulldowns occurred while rotating for 16 hr at 4°C. Beads were then washed with TSE I (20 mMTris/HCl pH 7.4 at 20°C, 150 mM NaCl, 0.1% SDS, 1% Triton X-100, 2 mM EDTA), twice with TSE III (10 mM Tris/HCl pH 7.4 at 20°C, 250 mM LiCl, 1% IGEPAL CA-630, 0.7% Deoxycholate, 1 mM EDTA), and twice with TE followed by elution from the beads using elution buffer (0.1 M NaHCO$_3$, 1% SDS). Elutions were subsequently de-crosslinked overnight at 65°C and DNA was purified using ChIP DNA Clean and Concentrator (Zymo Research) and DNA was used for qPCR. The primer sequences used in this study are:

*Cd69*_ChIP_F TCCCTGCTGTCTGAAATGTG
*Cd69*_ChIP_R GTGGAAGGATGTCTTCGATTCT
*Ifit2*_ChIP_F GCATTGTGCAAGGAGAATTCTATG
*Ifit2*_ChIP_R TTCCGGAATTGGGAGAGAGA
*Gbp5*_ChIP_F TAAACAGCGCTTGAAACAATGA
*Gbp5*_ChIP_R AGGCTTGAATGTCACTGAACTA

## Luciferase assay

Cells were plated in a 96-well plate and transfected when approximately 80% confluent. Transfections of pRL-CMV encoding Renilla luciferase (*Heinz et al., 2010*), together with a firefly luciferase promoter-reporter construct containing eight GAS consensus sequences (*Horvai et al., 1997*), control vector, or the indicated PHLPP constructs, were performed using Lipofectamine 3000 reagent (Invitrogen, L3000) for MEFs or Fugene six reagent (Promega, E269A) for HEK-293T cells. Cells were treated with murine or human IFNγ (PeproTech, 315–05, 300–02, respectively) for the indicated times at 37°C and activity was measured using the Dual-Glo Luciferase Assay System (Promega, E2940) in a Tecan Infinite M200 Pro multi-well plate reader. Promoter activity was corrected for the luciferase activity of the internal control plasmid, pRL-CMV, and Relative Response Ratios (RRR) were calculated.

## Immunoprecipitation and western blot

DNA was transfected into HEK-293T cells using FuGene 6. Cells were collected 24 hr post-transfection and then lysed in a buffer containing 50 mM Na$_2$HPO$_4$ (pH 7.5), 1 mM sodium pyrophosphate, 20 mM NaF, 2 mM EDTA, 2 mM EGTA, 1% SDS, 1 mM DTT, 1 μM microcystin, 20 μM benzamidine, 40 μg/ml leupeptin, and 1 mM PMSF and then were sonicated briefly. For co-immunoprecipitation, cells were lysed and the detergent-solubilized cell lysates were incubated with an anti-HA antibody (BioLegend, 901503) at 4°C overnight. Samples were incubated with protein A/G PLUS-Agarose (Santa Cruz Cat sc-2003) for 1 hr at 4°C and washed three times in lysis buffer containing 0.3 M NaCl and 0.1% Triton X 100. Bound proteins and lysates were separated by SDS/PAGE gel and analyzed by western blot.

## Immunofluorescence

HeLa cells were plated on glass coverslips and transfected using FuGene 6. 24 hr after transfection, cells were fixed with 4% paraformaldehyde for 20 min at room temperature, followed by fixation with 100% methanol for 3 min at $-20°C$. Cells were permeabilized and blocked in 0.3% Triton X 100% and 3% BSA for 30 min at room temperature, followed by three 5 min washes in PBS-T. Primary antibodies were diluted at the following dilutions: mouse anti-HA, 1:500; rabbit anti-$\alpha$-tubulin (Cell Signaling, 2125), 1:200. Secondary antibodies were diluted at the following dilutions: Alexa647 anti-mouse (Life Technologies, A21235), 1:500; Alexa488 anti-rabbit (Life Technologies, A11034), 1:500. Coverslips were mounted onto slides with ProLong Diamond Antifade Mountant with DAPI (ThermoFisher, P36966). Images were acquired on a Zeiss Axiovert 200M microscope (Carl Zeiss Microimaging Inc) using an iXon Ultra 888 EMCCD camera (ANDOR) controlled by MetaFluor software (Molecular Devices) and analyzed on ImageJ (NIH). The Nuclear to Cytoplasmic ratio was calculated as follows: the mean signal intensity was measured for a region of the nucleus and cytoplasm for each cell, and the mean signal intensity of the background was subtracted from these values. Then the Nuclear to Cytoplasmic ratio was calculated by dividing the background subtracted mean signal intensity for the nuclear signal by the background subtracted value for the cytoplasmic signal.

## *In vitro* phosphatase assay

pCMV 3xFLAG PHLPP1 was transfected into HEK-293T cells plated in four 10 cm plates (approximately $9 \times 10^6$ cells per plate, 80% transfection efficiency) using Fugene 6. Cells were collected 48 hr post-transfection and lysed in a buffer containing 20 mM Tris (pH 7.5), 150 mM NaCl, 1 mM EDTA, 1 mM EGTA, 1% Triton X 100, 2.5 mM sodium pyrophosphate, 1 mM $Na_3VO_4$, 1 mM DTT, 1 mM PMSF, 1 µM microcystin, 20 µM benzamidine, and 40 µg/ml leupeptin. The detergent-solubilized cell lysates were incubated with anti-FLAG M2 affinity gel (30 µl per plate, Sigma-Aldrich, A2220) for 1 hr at 4°C, washed four times in lysis buffer and the beads were resuspended in 40 µl 200 mM Tris, 4 mM DTT, 20 mM $MnCl_2$ for use in *in vitro* phosphatase assay. STAT1 (0.3 µM) (Biosource, PHF0011) was phosphorylated *in vitro* by incubation with recombinant human cdk1/cyclinB (0.2 µM) (Millipore, 14–450) at 30°C for 90 min in the presence of 1 mM ATP, and 1 X PK buffer (NEB, B6022) containing 50 mM Tris, 10 mM $MgCl_2$, 0.1 mM EDTA, 2 mM DTT, 0.01% Brij, pH 7.5, and the reaction was quenched by addition of 144 µM CDK1 inhibitor RO3306 (Enzo, ALX-270–463). Phosphorylated STAT1 substrate was added to 1/4 vol of beads with bound PHLPP1 (or to lysis buffer control) and reactions were allowed to proceed for an additional 120 min at 30°C. For the zero minute time point, beads were added after the 120 min incubation and all reactions were immediately quenched with 4xSB (sample buffer). Samples were analyzed by western blot.

## Mouse infection and endotoxin challenge

Bacterial sepsis in mice was induced by injection of *E. coli* K1 strain RS218 and LPS endotoxemia was induced by injection of purified *E. coli* O111:B4 LPS (Sigma-Aldrich). The *E. coli* culture was grown overnight in Luria broth (LB) medium (Hardy Diagnostics) at 37°C with shaking. The bacterial culture was diluted 1:50 in fresh LB, grown to mid-log phase, washed twice with PBS and reconstituted in PBS to yield the appropriate inoculum. For survival experiments, 10 to 14 week-old female C57BL/6 WT and littermate control *Phlpp1$^{-/-}$* mice were injected i.p. with $5 \times 10^7$ colony forming units (cfu) *E. coli* or 15 mg/kg LPS and mouse survival recorded for 10 days following injection. For measurement of serum IL-6, IL-10 and IL-1β levels, mice were injected with 10 mg/kg LPS, and at 4, 8, 12 and 24 hr after injection, 80 µl of blood was collected by submandibular bleeding using a lancet into a serum separating blood collection tubes (BD) that were spun at $1500 \times g$ for 10 min to separate serum. Serum cytokines were quantified by specific ELISA (R and D) following the manufacturer's protocol. All protocols for mouse experiments were conducted in accordance with the institutional guidelines and were approved by the Institutional Animal Care and Usage Committee (IACUC) at the University of California, San Diego.

## Acknowledgements

We thank Maya Kunkel for advice and Mira Sastri, Gregory Fonseca, and Ali Syed for assistance with reagents and methodologies. This work was supported by NIH R35 GM122523 (ACN), NIH

GM067946 (ACN), NIH HL125352 (VN), NIH DK091183 (CKG) and DK063491 (CKG). KC-K. was supported in part by NIH/NCI T32 CA009523 and ATK was supported in part by the University of California San Diego Graduate Training Grant in Cellular and Molecular Pharmacology through the National Institutes of Health Institutional Training Grant T32 GM007752 from the NIGMS.

## Additional information

### Funding

| Funder | Grant reference number | Author |
| --- | --- | --- |
| National Institutes of Health | R35 GM122523 | Alexandra C Newton |
| National Institutes of Health | DK091183 | Christopher K Glass |
| National Institutes of Health | GM067946 | Alexandra C Newton |
| National Institutes of Health | HL125352 | Victor Nizet |
| National Institutes of Health | DK063491 | Christopher K Glass |
| National Institutes of Health | T32 CA009523 | Ksenya Cohen Katsenelson |
| National Institutes of Health | T32 GM007752 | Agnieszka T Kawashima |

The funders had no role in study design, data collection and interpretation, or the decision to submit the work for publication.

### Author contributions

Ksenya Cohen Katsenelson, Conceptualization, Resources, Data curation, Formal analysis, Supervision, Funding acquisition, Validation, Investigation, Visualization, Methodology, Writing—original draft, Project administration, Writing—review and editing, Performed the experiments, Conceived the project; Joshua D Stender, Satoshi Uchiyama, Conceptualization, Data curation, Formal analysis, Validation, Investigation, Visualization, Methodology, Writing—original draft, Writing—review and editing, Performed the experiments, Conceived the project; Agnieszka T Kawashima, Gema Lordén, Conceptualization, Resources, Data curation, Formal analysis, Supervision, Investigation, Methodology, Writing—review and editing, Performed the experiments, Conceived the project; Victor Nizet, Christopher K Glass, Conceptualization, Resources, Data curation, Formal analysis, Supervision, Investigation, Methodology, Writing—review and editing, Conceived the project; Alexandra C Newton, Conceptualization, Resources, Data curation, Formal analysis, Supervision, Funding acquisition, Investigation, Methodology, Writing—original draft, Project administration, Writing—review and editing, Conceived the project

### Author ORCIDs

Agnieszka T Kawashima https://orcid.org/0000-0003-0176-1656
Satoshi Uchiyama https://orcid.org/0000-0002-0979-7998
Victor Nizet https://orcid.org/0000-0003-3847-0422
Christopher K Glass https://orcid.org/0000-0003-4344-3592
Alexandra C Newton https://orcid.org/0000-0002-2906-3705

### Ethics

Animal experimentation: All protocols for mouse experiments were conducted in accordance with the institutional guidelines and were approved by the Institutional Animal Care and Usage Committee (IACUC) at the University of California, San Diego. IACUC protocol numbers S06288, S00227M.

### Decision letter and Author response

Decision letter https://doi.org/10.7554/eLife.48609.020
Author response https://doi.org/10.7554/eLife.48609.021

# Additional files

### Supplementary files

• Supplementary file 1. List of 199 KLA-induced genes that are elevated in *Phlpp1*$^{-/-}$ BMDMs compared to WT cells.
DOI: https://doi.org/10.7554/eLife.48609.014

• Supplementary file 2. List of 144 KLA-induced genes that are reduced in *Phlpp1*$^{-/-}$ BMDMs compared to WT cells.
DOI: https://doi.org/10.7554/eLife.48609.015

• Transparent reporting form
DOI: https://doi.org/10.7554/eLife.48609.016

### Data availability

Sequencing data have been deposited in GEO under accession code GSE116314. All data generated or analysed during this study are included in the manuscript and supporting files.

The following dataset was generated:

| Author(s) | Year | Dataset title | Dataset URL | Database and Identifier |
|---|---|---|---|---|
| Cohen Katsenelson K, Stender JD, Glass CK, Newton AC | 2018 | Transcriptomic changes in wild-type and Phlpp1-/- mice following KLA stimulation | https://www.ncbi.nlm.nih.gov/geo/query/acc.cgi?acc=GSE116314 | NCBI Gene Expression Omnibus, GSE116314 |

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
