## [Decision Letter]

[Editors’ note: a previous version of this study was rejected after peer review, but the authors submitted for reconsideration. The first decision letter after peer review is shown below.]

Thank you for submitting your work entitled "PHLPP1 counter-regulates STAT1-mediated inflammatory signaling" for consideration by *eLife*. Your article has been reviewed by three peer reviewers, including Kim Orth as the Reviewing Editor and Reviewer #1, and the evaluation has been overseen by a Reviewing Editor and a Senior Editor.

Our decision has been reached after consultation between the reviewers. Based on these discussions and the individual reviews below, we regret to inform you that your work will not be considered further for publication in *eLife*. Although the work is seen as important, it is not a complete story and would require experiments that will take longer than the allowed time. If the author chooses to address the issues brought up by these reviewers, we encourage them to submit a new manuscript to *eLife* with this data and we will do our best to use the same editors and reviewers for the new submission.

*Reviewer #1:*

The authors report that the PHLPP1 phosphatase specifically targets phosphorylated Ser727 on STAT1 and thereby inhibits its activity as a transcription factor. This is an important finding because STAT1 is a transcriptional activator that mediates the cellular response to interferons, cytokines and growth factors. The authors begin by showing that *Phlpp1^-/-^* mice have improved survival during *E. coli* infection and LPS-induced toxicity, indicating an important role for the PHLPP1 phosphatase in innate immune signaling. The authors then performed RNA-seq BMDMs derived from WT and *Phlpp1^-/-^* mice to determine how loss of Phlpp1 altered gene expression and found a significant enrichment in STAT binding motifs in the Phlpp1-null elevated transcripts. Importantly, the authors validated the Phlpp1's effect on STAT1 promoter occupancy through ChIP experiments. Finally, the authors show that dephosphorylation of Ser727 on STAT1 does indeed suppress its activity as a transcription factor and that PHLPP1 specifically dephosphorylates Ser727 *in vitro* and that loss of Phlpp1 increases basal STAT1 Ser727 phosphorylation in vivo. Although this manuscript elucidates an important level of control in STAT1 signaling the following major and minor concerns must be addressed before this manuscript can be accepted for publication.

Essential revisions:

Figure 2E-G:

The text states that these are RT-PCR data (subsection “Loss of PHLPP1 results in a prolonged STAT1-dependent transcription in macrophages”), however the figure legend states that they are RNA-seq values. Which are they? If they are qRT-PCR then are they relative expression values or absolute expression? There is no detailed information in the Materials and methods section as to how the qPCR was performed and no sequences for primers to normalization genes if they are indeed relative values. The reference given is behind a paywall so the common reader would not be able to access any detailed information about the qRT-PCR methods used. The methods used to obtain these data must be further elucidated to be able to determine if the proper controls and normalization were performed.

Figure 5:

The analysis of the subcellular localization of PHLPP1 would be greatly improved by the addition of an experiment in which nuclear export was inhibited by Leptomcyin B. The authors reported that full-length PHLPP1 is cytosolic, but they failed to take into account that nuclear export/import is an equilibrium process and the only way to tell if a protein is indeed primarily localized in the cytosol is to block nuclear export.

It would also strengthen this line of inquiry if the authors tried (even if just with bioinformatics) to identify the native NLS of PHLPP1. The authors add a foreign NLS (from what protein? what type of NLS?) to show that driving the protein to the nucleus inhibits STAT1 promoter activity suggesting nuclear localization of PHLPP1 is important for its activity. They do not elucidate PHLPP1's native NLS, which must exist in order for the effect of nuclear localization of PHLPP1 on STAT1 promoter activity to actually be significant in vivo.

Figure 5D shows that PHLPP1 dephosphorylates STAT1 Ser727 *in vitro* and is out of place in Figure 5 which is all about nuclear localization. Figure 5D should either be incorporated into Figure 6 or moved to the supplement. This would leave room in Figure 5 for the suggested Leptomycin B experiment and NLS bioinformatics in this figure.

Figure 7:

Can you comment on the fact that pAkt T308 is elevated in all conditions in the PHLPP1 KO in 7A? PHLPP1 dephosphorylation of Ser473 was mentioned in the Discussion section, but not T308.

As it is presented, Figure 7C is confusing. It seems like the luciferase assay graph and the pAkt blot should be separated into Figure 7C and 7D or at least the blot needs to be mentioned in the figure legend which currently only explains the luciferase assay for Figure 7C. I understand that the blot is only for validation purposes, but it still needs to be mentioned in the figure legend if it is be included.

Subsection “PHLPP1 binds to STAT1 and dephosphorylates Ser727”: Please revise to "…interacts with STAT1 and reduces its promoter activity". You have shown that the PHLPP1 NTE can pull down STAT1 and that the PHLPP1 NTE reduces STAT1 promoter activity, but not exactly that one causes the other.

And as these were IPed out of lysate and not shown to directly interact with purified recombinant proteins, a direct interaction cannot be claimed.

*Reviewer #2:*

The presented paper identifies a potential phosphatase for STAT1. Deletion of the phosphatase-encoding gene Phlpp1 increases survival of mice upon LPS or *E. coli* challenge. Molecular analysis in macrophages revealed that PHLPP1 dephosphorylates Ser727 in STAT1 and thus suppresses inflammatory signaling.

Strong points of the paper are its detail in the characterization of PHLPP1 and its effect on STAT1. Weak points of the paper are the apparent presence of PHLPP1 even in knockouts, minimal in vivo analysis and exclusive focus on macrophages.

1) The manuscript is lacking controls and/or appropriate quantification for their knockdown and knockouts. The authors should show the efficiency of Phlpp1 knockdown by WB or PCR and that siCtl does not affect STAT1. Most importantly, neither the CRISPR knockout (Figure 6A) in BMDM nor BMDM from PHLPP1 KO (Figure S1) are convincingly lacking PHLPP1. The band intensity of PHLPP1 is almost the same between knockout and WT. Why is there a band at all for PHLPP1 in PHLPP1 KO? In Figure 6 and Figure S1, there seems to be only one band, whereas in Figure 7C the higher molecular band is marked with the arrow for PHLPP1. The authors have to show convincingly that PHLPP1 is absent in their KO cells.

2) The focus of the manuscript is the characterization of molecular interactions of PHLPP1. Nevertheless, the in vivo experiments should be analyzed in more detail. Is there increased dissemination in peripheral organs and blood in WT mice compared to KO mice? The increase in inflammatory cytokines could be either due to a higher inflammatory profile of immune cells or an increased recruitment/number. Thus, ideally the authors would measure recruitment of neutrophils, macrophages and DCs to the peritoneum and lymph nodes.

3) The authors show in vivo levels of only 3 cytokines and conclude from 2 cytokines that *Phlpp1^-/-^* mice had a diminished initial pro-inflammatory cytokine burst. Only IL-1β levels were consistently higher in KO mice. The authors should analyze more cytokines to draw this conclusion. For *in vitro* experiments, e.g. in vivo levels of IFNγ would be highly relevant and should be measured.

4) Figure 5C. Is the transfection efficiency the same? From the images, it seems that different constructs have different transfection efficacy. Confocal images lack scale bars. Bar graphs lack statistical analysis.

5) The rationale for choosing GBP5, IFIT2 and CD69 as representative genes should be provided.

*Reviewer #3:*

This is an interesting study that seeks to identify roles of the PHLPP1 phosphatase. The authors conclude that PHLPP1 dephosphorylates STAT1 on Ser727 and serves to promote resolution of inflammation by suppressing STAT1 activity. This is an important finding because while the kinases that cause STAT1 Ser727 phosphorylation have been characterized, little is known concerning the mechanism of dephosphorylation.

The authors' conclusions are supported by the data presented, although several aspects of the analysis appear incomplete.

1) The biological focus of the study is the protection against endotoxin and bacterial sepsis while the mechanistic studies focus on the resolution of inflammation. There is a logical problem here. Indeed, this is acknowledged by the authors who briefly state in subsection “PHLPP1 regulates the innate immune response” that the reduced mortality is most likely associated with a diminished initial cytokine response. The mechanism of the reduced initial cytokine response is not defined. Instead, the manuscript focusses on a second phenotype – reduced resolution – that may have little to do with the reduced mortality associated with endotoxin and bacterial sepsis. This lack of coherence between biology and mechanism needs to be fixed.

2) The conclusion that PHLPP1 deficiency promotes sustained STAT1 signaling is not strongly supported by the gene expression data (Figure 2E-G) that show increased expression at 6 hours and a similar level of decline in expression at 24 hours. Since the heat map is presented as the ratio of KO:WT, it is not clear that this expression pattern is not shared by the other genes in this group. A similar argument could be made for some of the genes in the ChIP analysis that show increased binding at all time points (Figure 3D-F). Please clarify. Moreover, this increase in gene expression at all time points correlates poorly with the decrease in the initial cytokine response that may contribute to the protection against sepsis and LPS (see #1 above).

3) The authors conclude (Abstract) that PHLPP1 catalytic activity is required for suppression of STAT1. The data to support this conclusion is incomplete. While the authors do show – by mutational analysis – that PHLPP1 catalytic activity is required for PHLPP1 Ser727 dephosphorylation *in vitro* (Figure 5D), this is not demonstrated in vivo. For example, knockout macrophages complemented with WT and catalytically inactive PHLPP1 should be compared to investigate the regulation of STAT1 Ser727 phosphorylation.

4) The CRISPR studies are unclear. First, the KO cells still express PHLPP1 (Figure 6A and Figure S1). Second, no controls for the CRISPR KO are presented. Third, the conclusion that KLA-induced phosphorylation is sustained at 24 hours in the PHLPP1 KO cells is not convincing (Figure S1). Fourth, it is unclear why *Phlpp1^-/-^* BMDM were not used for these studies – these cells prepared from *Phlpp1^-/-^* mice would not express PHLPP1 protein.

5) No information is provided concerning methods and statistical analysis of the promoter motif analysis (Subsection “Loss of PHLPP1 results in a prolonged STAT1-dependent transcription in macrophages” and Figure 2C,D legends).

6) No controls for the siRNA SMARTpool study (Figure 3C) are presented. Minimally, this requires the use of two independent single oligonucleotides. Preferably also complementation analysis.

[Editors’ note: what now follows is the decision letter after the authors submitted for further consideration.]

Thank you for submitting your article "PHLPP1 Counter-regulates STAT1-mediated Inflammatory Signaling" for consideration by *eLife*. Your article has been reviewed by three peer reviewers, including Kim Orth as the Reviewing Editor and Reviewer #1, and the evaluation has been overseen by a Reviewing Editor and Tadatsugu Taniguchi as the Senior Editor.

The reviewers have discussed the reviews with one another and the Reviewing Editor has drafted this decision to help you prepare a revised submission.

Summary:

The authors added significant amount of data for their molecular analysis of the role of PHLPP1 in suppressing STAT1 signaling.

Essential revisions:

*Reviewer #1:*

The authors have satisfied this reviewer with their revisions to the first submission.

*Reviewer #2:*

The authors added significant amount of data for their molecular analysis of the role of PHLPP1 in suppressing STAT1 signaling. Comments addressing the concerns of reviewer 1 and the use of BMDMs from *Phlpp1^-/-^* mice as suggested by reviewer 3 significantly improved the quality of the manuscript and support the claims made.

In their rebuttal, the authors wrote that they replaced all knockdown experiments with BMDMs. However, siRNA knockdowns of STAT1 are still used in Figure 3. Is there a reason for why the authors did not show that STAT1 is actually knocked down, that siCrtl does not affect STAT1, and the efficiency of the knockdown?

The in vivo role of PHLPP1 remains still largely unexplored. The authors acknowledge this and state that this is not in their area of expertise. The cytokine profiles are only measured until 24 hours post injection of LPS though, at which time the mortality rate between the two groups is identical. Difference are observed 48 hours post infection. As authors are unable to repeat the experiment and/or measure more cytokines, this should at least be clearly mentioned. The molecular analysis that the authors focus on seems sound, but the evidence that the observed in vivo phenotype is due to the lack of dephosphorylation of STAT1 is minimal. As the authors suggest exploration of PHLPP1 inhibitors for treatment of sepsis in the Discussion section, this lack of direct evidence should be mentioned.

*Reviewer #3:*

The manuscript has been improved. While the authors have not addressed all the points raised in my critique, I believe that the revised manuscript does address the primary criticisms and that the paper is now acceptable for publication.

---

## [Author Response]

[Editors’ note: the author responses to the first round of peer review follow.]

Reviewer #1:The authors report that the PHLPP1 phosphatase specifically targets phosphorylated Ser727 on STAT1 and thereby inhibits its activity as a transcription factor. This is an important finding because STAT1 is a transcriptional activator that mediates the cellular response to interferons, cytokines and growth factors. The authors begin by showing that Phlpp1-/- mice have improved survival during E. coli infection and LPS-induced toxicity, indicating an important role for the PHLPP1 phosphatase in innate immune signaling. The authors then performed RNA-seq BMDMs derived from WT and Phlpp1-/- mice to determine how loss of Phlpp1 altered gene expression and found a significant enrichment in STAT binding motifs in the Phlpp1-null elevated transcripts. Importantly, the authors validated the Phlpp1's effect on STAT1 promoter occupancy through ChIP experiments. Finally, the authors show that dephosphorylation of Ser727 on STAT1 does indeed suppress its activity as a transcription factor and that PHLPP1 specifically dephosphorylates Ser727 in vitro and that loss of Phlpp1 increases basal STAT1 Ser727 phosphorylation in vivo. Although this manuscript elucidates an important level of control in STAT1 signaling the following major and minor concerns must be addressed before this manuscript can be accepted for publication. Essential revisions: Figure 2E-G: The text states that these are RT-PCR data (subsection “Loss of PHLPP1 results in a prolonged STAT1-dependent transcription in macrophages”), however the figure legend states that they are RNA-seq values. Which are they? If they are qRT-PCR then are they relative expression values or absolute expression? There is no detailed information in the Materials and methods section as to how the qPCR was performed and no sequences for primers to normalization genes if they are indeed relative values. The reference given is behind a paywall so the common reader would not be able to access any detailed information about the qRT-PCR methods used. The methods used to obtain these data must be further elucidated to be able to determine if the proper controls and normalization were performed.

The data in Figure 2E-G are RNA-Seq, whereas those in Figure 3A-C are qPCR. We have updated the text to indicate “normalized mRNA-Seq data” where appropriate. In addition, we have updated the Materials and methods section to include the primer sequences for 36B4 that were used for normalization. We have also included additional references for the Materials and methods section.

Figure 5:The analysis of the subcellular localization of PHLPP1 would be greatly improved by the addition of an experiment in which nuclear export was inhibited by Leptomcyin B. The authors reported that full-length PHLPP1 is cytosolic, but they failed to take into account that nuclear export/import is an equilibrium process and the only way to tell if a protein is indeed primarily localized in the cytosol is to block nuclear export.It would also strengthen this line of inquiry if the authors tried (even if just with bioinformatics) to identify the native NLS of PHLPP1.

We have now identified a bipartite NLS in the unstructured N-Terminal Extension (NTE) of PHLPP1 (new Figure 6D). We show that a construct encoding the NTE localizes to the nucleus, that mutation of either half of the bipartite mutation significantly increases localization in the cytosol, and that mutation of both halves of the bipartite motif results in cytosolic localization that is almost as complete as the addition of a strong NES (new Figure 6E and 6F). Additionally, we show that the decrease in STAT1 promoter activity observed upon overexpression of PHLPP1 in HEK cells depends on a functional NLS: mutation of either or both halves of the NLS reduces or abolishes, respectively, the PHLPP1-dependent effects on STAT1 promoter activity (new Figure 6G). Additionally, we identify a nuclear exclusion signal (NES) preceding the phosphatase domain (Figure 6—figure supplement 2). We appreciate the reviewer’s suggestion we address the mechanism for nuclear localization as identification of the NLS provides the first functional role of the NTE, which is unique to PHLPP1. We have also revised the model in Figure 8 to indicate the NLS and NES.

The authors add a foreign NLS (from what protein? what type of NLS?) to show that driving the protein to the nucleus inhibits STAT1 promoter activity suggesting nuclear localization of PHLPP1 is important for its activity.

We have added the information on the NLS, which is from PKI.

They do not elucidate PHLPP1's native NLS, which must exist in order for the effect of nuclear localization of PHLPP1 on STAT1 promoter activity to actually be significant in vivo.

See above.

Figure 5D shows that PHLPP1 dephosphorylates STAT1 Ser727 in vitro and is out of place in Figure 5 which is all about nuclear localization. Figure 5D should either be incorporated into Figure 6 or moved to the supplement. This would leave room in Figure 5 for the suggested Leptomycin B experiment and NLS bioinformatics in this figure.

We appreciate these helpful recommendations and have reorganized all the biochemical figures as follows: the *in vitro* dephosphorylation assay (previously Figure 5D) is now presented in the new Figure 4, which focusses exclusively on phosphorylation of Ser727 of STAT1 by PHLPP1. Specifically, the figure shows that (1) STAT1 phosphorylation on Ser727 is enhanced in primary bone marrow derived macrophages from *Phlpp1^-/-^* mice compared to WT mice (new Figure 6 A,B), (2) pSer727 is dephosphorylated *in vitro* by PHLPP1 (Figure 4 C,D), and (3) STAT1 has reduced phosphorylation on Ser727 in cells overexpressing PHLPP1 (Figure 4 D,E).

Figure 7:Can you comment on the fact that pAkt T308 is elevated in all conditions in the PHLPP1 KO in 7A? PHLPP1 dephosphorylation of Ser473 was mentioned in the Discussion section, but not T308.

Phosphate at the hydrophobic motif (Ser473) decreases the phosphatase sensitivity at the activation loop (Thr308), that is why the PHLPP1 KO cells have elevated phosphorylation at both sites, even though only Ser473 is a direct PHLPP. In the revision, we replaced the original Figure 7A,B with a better analysis, which now is the new Figure 5A,B. Here, we make the point that the Ser727 phosphorylation is okadaic-acid insensitive in the primary bone marrow-derived macrophages, under conditions where known PPA sites are okadaic-acid sensitive (the phosphorylation of T308 on Akt and T202/Y204 on Erk increases with okadaic acid yet there is no detectable change on Ser727 phosphorylation).

As it is presented, Figure 7C is confusing. It seems like the luciferase assay graph and the pAkt blot should be separated into Figure 7C and 7D or at least the blot needs to be mentioned in the figure legend which currently only explains the luciferase assay for Figure 7C. I understand that the blot is only for validation purposes, but it still needs to be mentioned in the figure legend if it is be included.

The luciferase assay is now Figure 6A; we removed the blot and replaced it with the new data in the new Figure 5A,B.

Subsection “PHLPP1 binds to STAT1 and dephosphorylates Ser727”: Please revise to "…interacts with STAT1 and reduces its promoter activity". You have shown that the PHLPP1 NTE can pull down STAT1 and that the PHLPP1 NTE reduces STAT1 promoter activity, but not exactly that one causes the other.And as these were IPed out of lysate and not shown to directly interact with purified recombinant proteins, a direct interaction cannot be claimed.

Point well taken; text modified as suggested.

Reviewer #2:The presented paper identifies a potential phosphatase for STAT1. Deletion of the phosphatase-encoding gene Phlpp1 increases survival of mice upon LPS or E. coli challenge. Molecular analysis in macrophages revealed that PHLPP1 dephosphorylates Ser727 in STAT1 and thus suppresses inflammatory signaling.Strong points of the paper are its detail in the characterization of PHLPP1 and its effect on STAT1. Weak points of the paper are the apparent presence of PHLPP1 even in knockouts, minimal in vivo analysis and exclusive focus on macrophages.1) The manuscript is lacking controls and/or appropriate quantification for their knockdown and knockouts. The authors should show the efficiency of Phlpp1 knockdown by WB or PCR and that siCtl does not affect STAT1. Most importantly, neither the CRISPR knockout (Figure 6A) in BMDM nor BMDM from PHLPP1 KO (Figure S1) are convincingly lacking PHLPP1. The band intensity of PHLPP1 is almost the same between knockout and WT. Why is there a band at all for PHLPP1 in PHLPP1 KO?

We have replaced all the experiments using CRISPR knockdown with ones using primary BMDM isolated directly from the WT vs. *Phlpp1^-/-^* mice. The new Figure 4A compares the KLA-induced phosphorylation of STAT1 at Ser727 and Tyr701 in these macrophages, with the data from 5 independent experiments quantified in the new Figure 5B.

In Figure 6 and Figure S1, there seems to be only one band, whereas in Figure 7C the higher molecular band is marked with the arrow for PHLPP1. The authors have to show convincingly that PHLPP1 is absent in their KO cells.

As noted above, Figure 6 (with the CRIPSR knockdown) has been replaced with experiments using BMDM from the *Phlpp1^-/-^* mice. The immunoreactive band in the PHLPP1 blot in original Figure 7C (now Figure 4C) that is present in all lanes is β-catenin; all commercially available PHLPP1 antibodies raised to the C-term of PHLPP1 seem to label it (Lobert et al., 2013).

2) The focus of the manuscript is the characterization of molecular interactions of PHLPP1. Nevertheless, the in vivo experiments should be analyzed in more detail. Is there increased dissemination in peripheral organs and blood in WT mice compared to KO mice? The increase in inflammatory cytokines could be either due to a higher inflammatory profile of immune cells or an increased recruitment/number. Thus, ideally the authors would measure recruitment of neutrophils, macrophages and DCs to the peritoneum and lymph nodes.

These are very interesting suggestions but beyond the scope of our expertise. Additionally, the two co-first authors are no longer in our labs, and the third author is on maternity leave. While we recruited another postdoc to address all the biochemical revisions, we are not equipped to do whole animal studies. Thus, although very interesting experiments, our current contribution focuses on the molecular mechanisms of how PHLPP1 suppresses STAT1 signaling.

3) The authors show in vivo levels of only 3 cytokines and conclude from 2 cytokines that Phlpp1^-/-^ mice had a diminished initial pro-inflammatory cytokine burst. Only IL-1β levels were consistently higher in KO mice. The authors should analyze more cytokines to draw this conclusion. For in vitro experiments, e.g. in vivo levels of IFNγ would be highly relevant and should be measured.

We have focused on the mechanistic aspects given our lack of personnel (please see previous comment).

4) Figure 5C. Is the transfection efficiency the same? From the images, it seems that different constructs have different transfection efficacy. Confocal images lack scale bars. Bar graphs lack statistical analysis.

The transfection efficiency was similar for all constructs and ranged from 50-75%. We have now added a scale bar (15 µm) and statistical analysis.

5) The rationale for choosing GBP5, IFIT2 and CD69 as representative genes should be provided.

We chose these genes because they were significantly elevated in the *Phlpp1^-/-^* comparted to WT MEFs. Additionally, they have proximal STAT1 binding motifs in their promoter regions. We have clarified this in the text:

“We next selected three genes whose expression was elevated in the *Phlpp1^-/-^* compared to WT macrophages and which had proximal STAT1 binding motifs in their promoters to further analyze: *Cd69, Ifit2*, and *Gbp5*.”

Reviewer #3:This is an interesting study that seeks to identify roles of the PHLPP1 phosphatase. The authors conclude that PHLPP1 dephosphorylates STAT1 on Ser727 and serves to promote resolution of inflammation by suppressing STAT1 activity. This is an important finding because while the kinases that cause STAT1 Ser727 phosphorylation have been characterized, little is known concerning the mechanism of dephosphorylation.The authors' conclusions are supported by the data presented, although several aspects of the analysis appear incomplete.1) The biological focus of the study is the protection against endotoxin and bacterial sepsis while the mechanistic studies focus on the resolution of inflammation. There is a logical problem here. Indeed, this is acknowledged by the authors who briefly state in subsection “PHLPP1 regulates the innate immune response” that the reduced mortality is most likely associated with a diminished initial cytokine response. The mechanism of the reduced initial cytokine response is not defined. Instead, the manuscript focusses on a second phenotype – reduced resolution – that may have little to do with the reduced mortality associated with endotoxin and bacterial sepsis. This lack of coherence between biology and mechanism needs to be fixed.

The biological phenotype of the *Phlpp1^-/-^* mice likely arises from an abundance of perturbations to the normal physiology of the mouse. We begin with the mouse experiment because it clearly points to a defect in innate immunity, however the primary focus of our manuscript is to identify a previously undescribed mechanism for the suppression of STAT1 signaling by PHLPP1. This mechanism is likely one of many that contribute to the whole animal phenotype, and we have tried to express this in the manuscript.

2) The conclusion that PHLPP1 deficiency promotes sustained STAT1 signaling is not strongly supported by the gene expression data (Figure 2E-G) that show increased expression at 6 hours and a similar level of decline in expression at 24 hours. Since the heat map is presented as the ratio of KO:WT, it is not clear that this expression pattern is not shared by the other genes in this group. A similar argument could be made for some of the genes in the ChIP analysis that show increased binding at all time points (Figure 3D-F). Please clarify. Moreover, this increase in gene expression at all time points correlates poorly with the decrease in the initial cytokine response that may contribute to the protection against sepsis and LPS (see #1 above).

We acknowledge that the genes evaluated show a decline in expression at 24 hours, however they are expressed at significantly higher levels in the *Phlpp1^-/-^* macrophages compared with WT. We conclude that STAT1 signaling is sustained (or elevated) in the *Phlpp1^-/-^* macrophages relative to the WT macrophages based on the following observations:

1) There are elevated genes in the *Phlpp1^-/-^* relative to WT macrophages.

2) The promoter regions of these genes have an enrichment of STAT1 binding motifs relative to genome background.

3) The ChIP signal for STAT1 is elevated on representative promoters in *Phlpp1^-/-^* macrophages relative to WT cells.

4) STAT1 phosphorylation is enhanced and sustained in Phlpp1 KO macrophages relative to WT.

3) The authors conclude (Abstract) that PHLPP1 catalytic activity is required for suppression of STAT1. The data to support this conclusion is incomplete. While the authors do show – by mutational analysis – that PHLPP1 catalytic activity is required for PHLPP1 Ser727 dephosphorylation in vitro (Figure 5D), this is not demonstrated in vivo. For example, knockout macrophages complemented with WT and catalytically inactive PHLPP1 should be compared to investigate the regulation of STAT1 Ser727 phosphorylation.

Our new experiments focus on primary BMDM which are have exceptionally low transfection efficiency and not amenable to the rescue experiment noted. When we overexpress a catalytically-dead PHLPP1 PP2C domain, it no longer regulates STAT1 transcriptional activity whereas an active PP2C does. These cell-based assays are consistent with PHLPP1 regulating STAT1 via a catalytic mechanism. Additionally, PHLPP1 dephosphorylates Ser727 *in vitro*.

4) The CRISPR studies are unclear. First, the KO cells still express PHLPP1 (Figure 6A and Figure S1). Second, no controls for the CRISPR KO are presented. Third, the conclusion that KLA-induced phosphorylation is sustained at 24 hours in the PHLPP1 KO cells is not convincing (Figure S1). Fourth, it is unclear why Phlpp1^-/-^ BMDM were not used for these studies – these cells prepared from Phlpp1^-/-^ mice would not express PHLPP1 protein.

The reviewer brings up an excellent point, to use the *Phlpp1^-/-^* BMDM. We have replaced all the CRISPR KO with experiments using the BMDM; see new Figure 4A,B and new Figure 5A,B.

5) No information is provided concerning methods and statistical analysis of the promoter motif analysis (Subsection “Loss of PHLPP1 results in a prolonged STAT1-dependent transcription in macrophages” and Figure 2C,D legends).

We have updated the text to include more details on the promoter analysis; we have also updated the figure legends.

6) No controls for the siRNA SMARTpool study (Figure 3C) are presented. Minimally, this requires the use of two independent single oligonucleotides. Preferably also complementation analysis.

These studies were internally controlled for with a control siRNA and a SMARTpool of four oligos synthesized by Dharmacon. The purpose of this experiment was to demonstrate that these are targets of STAT1, and the presented data support this conclusion.

[Editors' note: the author responses to the re-review follow.]

Essential revisions:Reviewer #2:The authors added significant amount of data for their molecular analysis of the role of PHLPP1 in suppressing STAT1 signaling. Comments addressing the concerns of reviewer 1 and the use of BMDMs from Phlpp1^-/-^ mice as suggested by reviewer 3 significantly improved the quality of the manuscript and support the claims made.In their rebuttal, the authors wrote that they replaced all knockdown experiments with BMDMs. However, siRNA knockdowns of STAT1 are still used in Figure 3. Is there a reason for why the authors did not show that STAT1 is actually knocked down, that siCrtl does not affect STAT1, and the efficiency of the knockdown?

Apologies for the misleading statement – we replaced all the biochemical/mechanistic experiments addressing PHLPP1 loss with ones using BMDMS, but the reviewer is correct in noting that the validation experiments in Figure 3 were with siRNA knockdown of STAT1. The knockdown of STAT1 was confirmed by the RNA-Seq. Also, given the clear effect of the STAT1 knockdown, but not siCtrl, on genes with STAT1 promoters, we did not feel that addressing efficiency for these experiments was necessary.

The in vivo role of PHLPP1 remains still largely unexplored, which you acknowledge, stating that this is not in their area of expertise. The cytokine profiles are only measured until 24 hours post injection of LPS though, at which time the mortality rate between the two groups is identical. Differences are observed at 48 hours post infection. As you cannot repeat the experiment and/or measure more cytokines, this should at least be clearly mentioned. The molecular analysis that you focus on appears sound, but the evidence that the observed in vivo phenotype is due to the lack of dephosphorylation of STAT1 is minimal. As you suggest exploration of PHLPP1 inhibitors for treatment of sepsis in the Discussion section, this lack of direct evidence must be mentioned.

These are all good points and we have adjusted the text as in the Results section:

“Note that cytokine levels were measured up to 24 h post LPS injection, when the protective effect of PHLPP1 loss was not yet apparent.”

and in the Discussion section:

"Although the current study does not provide direct evidence that enhanced phosphorylation of STAT1 causes the protective effect of PHLPP1 loss on both *E. coli*-induced sepsis and LPS-induced endotoxemia in mice, our data indicate that PHLPP1 inhibitors could be explored as adjunctive therapies to antibiotics and supportive care of patients with Gram-negative sepsis, a leading cause of mortality in intensive care units.”